# GRACE constraints on Earth rheology of the Barents Sea and Fennoscandia

Marc Rovira-Navarro[1,2], Wouter van der Wal[1,3], Valentina R. Barletta[4], Bart C. Root[1], and Louise Sandberg Sørensen[4]

[1]TU Delft, Faculty of Aerospace Engineering, Building 62 Kluyverweg 1, 2629 HS Delft, The Netherlands
[2]NIOZ Royal Netherlands Institute for Sea Research, Department of Estuarine and Delta Systems EDS, and Utrecht University, P.O. Box 140, 4400 AC Yerseke, the Netherlands
[3]TU Delft, Faculty of Civil Engineering and Geosciences, Stevinweg 1, 2628 CN Delft, The Netherlands
[4]National Space Institute, DTU Space, Technical University of Denmark, Elektrovej Bygning 327, 2800 Kongens Lyngby, Denmark

**Correspondence:** Marc Rovira-Navarro (marc.rovira@nioz.nl)

**Abstract.** The Barents Sea is situated on a continental margin and was home to a large ice sheet at the Last Glacial Maximum. Studying the solid Earth response to the removal of this ice sheet (Glacial Isostatic Adjustment, GIA) can give in-sight to the sub-surface rheology of this region. However, because the region is currently covered by ocean, uplift measurements from the center of the former ice sheet are not available. The Gravity Recovery and Climate Experiment (GRACE) gravity data has been
5    shown to be able to constrain GIA. Here we analyze GRACE data for the period $2003 - 2015$ in the Barents Sea and use it to constrain GIA models for the region. We study the effect of uncertainty in non-tidal ocean mass models that are used to correct GRACE data and find that it should be taken into account when studying solid Earth signals in oceanic areas from GRACE. We compare GRACE derived gravity disturbance rates with GIA model predictions for different ice deglaciation chronologies of the last glacial cycle and find that best fitting models have an upper mantle viscosity equal or higher than $3 \cdot 10^{20} \ \mathrm{Pa \cdot s}$.
10   Following a similar procedure for Fennoscandia we find that the preferred upper mantle viscosity there is a factor 2 larger than in the Barents Sea for a range of lithospheric thickness values. This factor is shown to be consistent with the ratio of viscosities derived for both regions from global seismic models. The viscosity difference can serve as constraint for geodynamic models of the area.

## 1   Introduction

15   Ongoing viscous rebound of the solid Earth (Glacial Isostatic Adjustment, GIA) after the collapse of large ice sheets results in positive gravity disturbance rates in several regions of the Earth. The Gravity Recovery and Climate Experiment (GRACE) satellite data has been used to constrain numerical models for GIA in North America (Tamisiea et al., 2007; Paulson et al., 2007; van der Wal et al., 2008; Sasgen et al., 2012) and Fennoscandia (Steffen and Denker, 2008; van der Wal et al., 2011;

Simon et al., 2018). With longer time series it is now possible to observe weaker GIA signals such as that of the Svalbard-Barents-Kara Ice Sheet (SBKIS) in GRACE gravity data (Root et al., 2015a; Kachuck and Cathles, 2018; Simon et al., 2018). The use of GRACE data is especially relevant in this region as other geodetic observations normally used for GIA studies are only available from the islands surrounding the Barents Sea; in the periphery of the ice sheet that covered the region during the

Last Glacial Maximum (LGM). This makes GIA-based ice sheet reconstructions such as ICE-5G and ICE-6G (Peltier, 2004; Peltier et al., 2015; Argus et al., 2014) uncertain.

Earlier work on the SBKIS proposed the existence of an extensive ice sheet spanning from the British Islands to the Kara Sea and extending further into mainland Russia (e.g., Grosswald, 1980, 1998) but more recent studies favour a smaller ice sheet (e.g., Lambeck, 1995; Siegert and Dowdeswell, 1995; Svendsen et al., 1999, 2004; Mangerud et al., 2002). During the

last decade, more geological and glaciological observations relevant for reconstructing the SBKIS have been obtained and compiled in the first version of the DATabase of Eurasian Deglaciation (DATED-1) resulting in new ice sheet limits for the whole Eurasian Ice Sheet Complex (EISC) (Hughes et al., 2016), but ice thickness variations can not be uniquely established.

Comparing the GRACE-derived gravity disturbance rates with those predicted for different palaeo-ice sheet configurations, Root et al. (2015a) conclude that the SBKIS contained less ice than previously thought. Kachuck and Cathles (2018) use

GRACE data, along with Relative Sea Level (RSL) curves and GPS uplift measurements, to distinguish between two deglaciation histories: one with an ice sheet with a central dome in the Barents Sea and one with the Barents Sea marginally glaciated and domes in the surrounding Arctic islands. They show that the data is inconclusive in this regard.

Since the gravity disturbance rate signal in the Barents Sea region is small, it is important to thoroughly analyze the uncertainty in GRACE data. Here we present an extended analysis of GRACE data in the region and the different uncertainty

sources. We focus on the gravity disturbance rate due to non-tidal mass variations in the ocean which influence the secular signal from GRACE data in oceanic areas (de Linage et al., 2009). In the processing chain to obtain Level 2 GRACE data, changes in ocean-bottom pressure are removed using the Ocean Model for Circulation and Tides (OMCT) forced with atmospheric data from the European Centre for Medium-Range Weather Forecasts (ECMWF). However, the OMCT secular signal is not reliable and should not be interpreted geophysically (Dobslaw et al., 2013). Lemoine et al. (2007) use a different ocean

model in their GRACE data processing and find significant differences in the southern Arctic ocean.

We compare GRACE derived gravity disturbance rates to GIA model output to constrain the input of the GIA model. Because of uncertainty in solid Earth parameters and deglaciation history, it is difficult to uniquely constrain both. However, we can compare the GIA models for the Barents and Kara Sea areas with models for Fennoscandia constrained by the same data. In this way we can determine if there is a difference in Earth properties for both regions that is systematic for all deglaciation

chronologies. Such constraints on variation in viscosity are useful for GIA modelling and geodynamic modelling in general, as viscosity maps derived from laboratory experiments and seismic velocities are not sufficiently constrained (e.g., Barnhoorn et al., 2011). Furthermore, the Barents Sea is located on a continental margin, and knowledge of the subsurface rheology can help decipher its tectonic history. Our aim is to provide a constraint on upper mantle viscosity for the Barents Sea region and Fennoscandia, focusing on the difference in viscosity between the two regions. We build on existing knowledge of Earth

rheology and ice histories, which will be briefly reviewed in the following.

The rheology of the Barents Sea region is expected to be different from that of Fennoscandia, as it borders passive oceanic margins in the north and the west. Seismic tomography reveals lower seismic velocities in Barents Sea than below Fennoscandia (Levshin et al., 2007; Schaeffer and Lebedev, 2013), but not for all seismic periods and depths. 3D viscosity has been implemented in GIA models for the regions and has been found to affect sea level and uplift rates (Kaufmann and Wu, 1998).

However, the difference in properties between Fennoscandia and Barents Sea has not been studied explicitly.

Constraints from palaeoshoreline data on 1D GIA models resulted in best fitting upper mantle viscosities of $2-6 \cdot 10^{20} Pa \cdot s$ in the Barents Sea region (Steffen and Kaufmann, 2005), while recent work based on RSL data find that best fitting upper mantle viscosity in the Barents Sea region is above $2 \cdot 10^{20} Pa \cdot s$ (Auriac et al., 2016). For Fennoscandia, the best fitting upper mantle viscosity is found to be between $3-7 \cdot 10^{20} Pa \cdot s$ based on RSL data and relaxation time spectra, while best

fitting models based on GPS uplift rate measurements have upper mantle viscosities up to $15 \cdot 10^{20} Pa \cdot s$, see the overview in Steffen and Wu (2011). More recent work summarized in Simon et al. (2018) shows an upper mantle viscosity in the range of $3.4-20 \cdot 10^{20} Pa \cdot s$. Note that the lower bound for upper mantle viscosity in the Barents Sea is somewhat below that in Fennoscandia. Steffen and Kaufmann (2005) computed RSL misfit and find similar upper mantle viscosity for the Barents Sea and the Scandinavian mainland, but smaller lower mantle viscosity. However, the different studies used different ice histories

and relied on multiple data sources, with substantially less coverage in the Barents Sea region. Therefore it is unknown if it can be concluded from previous 1D studies whether viscosity is indeed lower in the Barents Sea than in Fennoscandia.

In this study we analyze GRACE data in the Barents Sea region and Fennoscandia to obtain the GIA signal there, focusing on the first region where the signal to noise ratio is lower. We compare the estimated signal with 1D GIA model output to infer upper or lower bounds in viscosity for different ice deglaciation chronologies. From comparison between the best fitting

models for the two regions we draw conclusions on the variation in Earth rheology between the Barents Sea and Fennoscandia.

## 2   Methodology

### 2.1   GRACE Data Processing

Temporal variations of the Earth's gravity field measured by GRACE are related to mass transport within the Earth system due to different geophysical processes, such as hydrology, ongoing cryospheric mass changes, GIA and (post-) seismic signals

(e.g., Wouters et al., 2014). To study GIA, other geophysical signals that mask the GIA signal should be removed. Additionally, GRACE data is affected by instrumental noise and the anisotropic sampling of the signal due to the satellites' orbit (Wahr, 2007; Flechtner et al., 2016). Different data-processing techniques have been developed to increase the GRACE signal-to-noise ratio (e.g., Han et al., 2005; Swenson and Wahr, 2006; Kusche et al., 2009). In the following, we detail the post-processing used to analyze GRACE data in the Barents Sea and Fennoscandia with focus on the Barents Sea as it presents additional difficulties

due to the smaller magnitude of the signal.

In our analysis we use the University of Texas Center for Space Research (UTCSR) release 5 (RL05) (Bettadpur, 2012) up to spherical harmonic degree 60. The difference between GRACE solutions up to degree 96 and degree 60 is shown to mainly manifest as north-south oriented stripes characteristic of the high-frequency noise in Grace, with a magnitude at the noise level

(Sakumura, 2014), therefore we do not use the coefficients beyond degree 60. Furthermore, their influence would be reduced because of the filtering that is applied in our processing as explained later in this section. We use data for the $2003 - 2015$ period. We substitute the degree two coefficients with those obtained from satellite laser ranging (Cheng et al., 2013). We use the least squares method to obtain the secular, annual and semiannual signals of each time series of Stokes' coefficients.

We estimate GRACE measurement errors ($\sigma_{GRACE}$) using the residuals after the secular, annual and semiannual signals are removed from the signal (Wahr et al., 2006).

After processing the signal as explained above, the GIA signal is evident as a positive gravity rate in Fennoscandia and the Barents Sea (Figure 1a). However, the signal is contaminated by the correlated noise in the higher degree GRACE data which is evident as north-south oriented stripes. We use a Gaussian filter to filter out short wavelength noise and apply the same filters

and maximum spherical harmonic degree to the GIA model output. The choice for filter half width affects the signal-to-noise ratio. Earlier studies in Scandinavia used a 400 km half width Gaussian filter (e.g. Steffen and Denker (2008), van der Wal et al. (2011)), but the better accuracy of later GRACE data releases and longer time series since then allows less filtering. To account for the fact that the optimum filter half width is not known, we adopt a range of high-pass filter half widths from 200 km to 300 km. At 200 km half width correlated noise (stripes) are still visible (Figure 1a,b), while for filter half widths

larger than 300 km the positive gravity anomaly in the Barents Sea is very small (Figure 2). Low-pass filtering to reduce the measurement noise inevitably means that some sensitivity to possible high-frequency signal content in the GIA models is lost, that is, we can not assess detailed changes in ice thickness based on our GRACE gravity rates. We additionally use a high-pass filter in the Barents Sea to remove the long-wavelength signal that contains unmodelled long wavelength phenomena such as global sea-level rise. The high-pass filter half width ranges from 500 to 700 km which was found to be optimum to remove

long wavelength signals while retaining most of the SBKIS GIA signal (see Root et al. (2015a) Supplementary Material). As the signal in Fennoscandia is larger and has a larger wavelength we only use the low-pass filter there with a half width that also ranges from 200 to 300 km.

The total observed gravity signal (Figures 1 a,b,d) cannot be directly interpreted as the GIA footprint of the paleo-ice sheet as it contains the trend of other geophysical processes as well, one of them being hydrology. Secular changes in land water storage

result in gravity trends that should be subtracted when analyzing GRACE data in continental areas. The long-term hydrology signal in Fennoscandia is probably small, as demonstrated by the good agreement between GIA signal derived from GRACE and GPS (van der Wal et al., 2011). However, the hydrology signal of the Russian Arctic Archipelago (Novaya Zemlya, Franz Josef Land and Severnaya Zemlya) can leak into oceanic areas. We subtract the hydrology signal using the GLDAS hydrology model (Rodell et al., 2004). Because its reliability for the islands of the Arctic Archipelago is not well-known, we follow

Matsuo and Heki (2013) and take the amplitude of its trend in the Barents Sea as an indication of the uncertainty in the hydrology signal in these polar regions ($\sigma_{hydrology}$).

Present-day changes in the cryosphere and the resulting present-day solid Earth response can also mask the GIA signal. In particular, the glaciers of the islands Svalbard and the Russian Arctic Archipelago are experiencing significant mass changes evident in GRACE observations which partly mask the GIA signal in the Barents Sea region (see Figure 1). Independent data

on mass changes in Svalbard and the Russian Arctic Archipelago is limited. Moholdt et al. (2012) derived trends using ICESat

for the 2003-2009 period using altimetry; other authors (e.g., Schrama et al., 2014; Matsuo and Heki, 2013) have used GRACE data. For the period 2003-2008, GRACE estimates are lower than altimetry estimates but agree within uncertainty (Root et al., 2015a). In Simon et al. (2018) ice mass loss estimates from altimetry and glaciology for a longer period were shown to be much larger than GRACE estimates in Svalbard, Franz Josef Land and Novaya Zemlya, and the former were scaled down in that study. Here we follow Root et al. (2015a) and use ice loss corrections obtained using the mascon method of Schrama et al. (2014) (see Table 1) to remove the ice loss signal taking into account elastic loading (Wahr et al., 1998).

To obtain the present-day mass changes from GRACE, a GIA correction needs to be first applied As our aim is to quantify the GIA signal in the central Barents Sea, the problem seems circular. However, the GIA model has a relatively small effect on the derived present-day mass changes. We account for uncertainty in mass loss estimations due to GIA by employing an ensemble of ice deglaciation chronologies and Earth rheological parameters. We use the ICE-5G model and two runs of the GSM ice model (Tarasov et al., 2012) with maximum and minimum ice sheet extents combined with the VM5a Earth model (Peltier, 2004) and an Earth model with a stronger mantle, as well as the W12 ice model (Whitehouse et al., 2012) with a strong mantle. Mass loss changes obtained using the different GIA models are shown in Table 1: more massive ice sheet models and stronger mantles result in higher mass loss rates. The error in the derived mass changes due to uncertainty in GIA is similar to the GRACE measurement error. We use the error bars of the estimated mass changes for Svalbard and the Russian Arctic Archipelago to estimate the error in the recovered GIA gravity rates due to uncertainty in mass loss changes in the region ($\sigma_{ice}$). Finally, for the Barents Sea, the Greenland mass loss is already filtered when using the high-pass filter, but for Fennoscandia we need to remove it. To do so we use ICESat mass changes from Sørensen et al. (2011).

We account for the uncertainty in non-tidal ocean changes by using the ECCO ocean model (Forget et al., 2015) as alternative for the ocean model used in standard GRACE level 2 processing. In that case we first add back the GAB products to restore the full GRACE ocean mass signal (Flechtner et al., 2015; Yu et al., 2018) before subtracting the ECCO ocean model. The ECCO model is a dynamically consistent ocean model constrained with observations from altimetry, Argo floats and GRACE. The model has been shown to correctly capture long-term bottom pressure variability in the Arctic Ocean and Adjacent Seas (Peralta-Ferriz, 2012). The version of the ocean model we use is the ECCOv4-llc270 compilation. This compilation covers the period 2001-2015 which means the GRACE time-series that we use in the Barents Sea is limited to this period. We obtain gravity rates in the central Barents Sea using the UTCSR GRACE solution corrected with both the OMCT and the ECCO ocean models. The differences between these two solutions are used as an indication of the uncertainty in non-tidal ocean changes ($\sigma_{ocean}$).

We estimate the total error in the gravity trends by assuming that the different error sources are uncorrelated:

$$\sigma = \sqrt{\sigma_{ice}^2 + \sigma_{GRACE}^2 + \sigma_{ocean}^2 + \sigma_{hydrology}^2}. \tag{1}$$

The assumption that errors are uncorrelated requires further discussion. GRACE data is assimilated in the ECCO ocean model. However, GRACE is only one of the 40 data sets used in the inversion process and the final product does not fit GRACE data well (Yu et al., 2018). Therefore there will be only a weak correlation with the GRACE data used in our estimation. Correlation

between land surface hydrology models and present-day ice melt is not expected, because hydrology models have little skill in predicting trends and do not model areas of permanent snow. Finally, ice loss changes errors ($\sigma_{ice}$) arise due to uncertainty in the GIA model and GRACE measurement error, we cannot rule out that the second error component might be correlated with $\sigma_{GRACE}$.

For the Barents Sea we consider the four terms; while for Fennoscandia we only consider GRACE measurement errors, as the ice loss changes in the Arctic Archipelago and ocean bottom pressure changes have a very small effect on the gravity trends recovered in Fennoscandia, and Greenland's mass loss signal is well known from altimetry measurements.

## 2.2  GIA Modelling

We compare GRACE derived gravity rates with those predicted by GIA models. To compute the gravity trends the sea level
equation is solved, using the pseudo-spectral approach presented in Mitrovica and Peltier (1991). We use the same code as Barletta and Bordoni (2013). To be able to run calculations for many different Earth parameters and ice models we assume that solid Earth properties only vary radially, which allows to compute GIA response for different regions separately with different viscosity profiles, but neglects effects of viscosity changes in surrounding regions. While this approach has been used in other GIA studies (Lambeck et al., 1998; Steffen et al., 2014), it has been suggested that far-field viscosity variations are relevant in
Fennoscandia (Whitehouse et al., 2006).

We neglect the loading effect due to sediment transport during deglaciation, as the effect is small and well below that of the unknown ice thickness (0.01 to 0.05 µGal/yr in Fennoscandia, and below 0.014 µGal/yr in the Barents Sea as shown in van der Wal and IJpelaar (2017) ). To study the effect of the ice deglaciation history on the present gravity rates we start by using a reference Earth model based on the averaged VM2 model which is similar to the VM5a model (Peltier, 2004; Argus
et al., 2014). The model consists of a 90 km lithosphere, a 570 km upper mantle with a viscosity of $0.5 \cdot 10^{21}$ Pa·s and a 2216 km lower mantle with an average viscosity of $2.6 \cdot 10^{21}$ Pa·s. The elastic properties of the Earth are based on the PREM model (Dziewonski and Anderson, 1981). To investigate the effect of the Earth's rheology, we vary the upper mantle viscosity between $0.1-1.6 \cdot 10^{21}$ Pa·s and the lithospheric thickness between $40-180$ km (Table 2). We do not change the lower mantle as its viscosity cannot be constrained uniquely from data in Fennoscandia (Steffen and Kaufmann, 2005). However, gravity
rates are influenced by the lower mantle viscosity which is discussed in the comparison with other GIA studies for the region.

We use an ensemble of ice histories that reflects the uncertainty in the deglaciation history of the European Ice Sheet Complex (EISC), the amount of ice in the SBKIS and the Scandinavian Ice Sheet (SIS) for the different ice deglaciation scenarios is shown in Figure 2 . The ice sheet models that we use can be divided in two main categories: (1) empirical ice sheet models based on GIA observables and empirically-determined ice extents, and (2) those based on numerical ice-sheet modeling forced
under different palaeo-climate scenarios and tuned to fit different constrains. A fundamental difference between these two kinds of models is that GIA-based palaeo-ice sheet models are explicitly associated with a specific Earth model. The first set of models is represented by the ICE-5G and ICE-6G models (Peltier, 2004; Peltier et al., 2015; Argus et al., 2014). Both models start the ice build-up 122 ka BP. The second set consists of three models obtained using the Glacial System Model (GSM) for Northern Europe (Tarasov et al., 2012), the University of Tromsø Ice Sheet Model (UiT ISM) (Patton et al., 2016, 2017), and

the S04 ice sheet model (Siegert and Dowdeswell, 2004), which are further described below. Figure 2 shows the ice model with the largest LGM ice volume (ICE-6G) and the model with the smallest ice volume (S04).

The three ice sheet models obtained using the GSM model are a subset of a bigger ensemble used in Root et al. (2015a) which showed good agreement with GRACE observations. The ensemble was obtained using a Bayesian calibration of GSM runs with RSL curves, present-day ground velocities and ice deglaciation margins from the DATED-1 project (Hughes et al., 2016). The VM5a rheology model was employed as reference during the calibration process, however, errors introduced by the rheology model were accounted for during the calibration process which implies that this model is not as strongly biased by a single viscosity profile as the ICE-5G and ICE-6G models. The three selected models consist of a late deglaciation model, labeled nn45283 in Root et al. (2015a) and two early deglaciation models, nn56536 and nn56597, with different maximum ice volumes. The build-up phase is faster than for the ICE-5G and ICE-6G models, build-up starts $28 \, \mathrm{ka}$ BP. The models will be labeled T1 (nn45283), T2 (nn56536) and T3 (nn56597) to simplify the notation.

The University of Tromsø Ice Sheet Model is based on a 3D thermomechanical ice model which uses an approximation of the Stokes equations forced by climatic and eustatic sea level perturbations to simulate the evolution of the EISC. The model is constrained using different geophysical and geological data sets including geomorphological flow sets, moraine and grounding zone wedge positions and isostasy patterns and is consistent with the DATED-1 ice sheet margins. Isostatic loading is implemented using the elastic lithosphere/ relaxed astenosphere model of Le Meur and Huybrechts (1996). The model has no ice in the region before $37 \, \mathrm{ka}$ BP.

Finally, we consider an ice sheet model which gives a lower bound for the mass present in the Barents Sea during the LGM, the S04 model (Siegert and Dowdeswell, 2004). The model is based on the continuity flow equations coupled with a model of water, basal sediment deformation and transportation. The model is forced with eustatic sea level curves of the last $30 \, \mathrm{ka}$ and palaeo air temperatures and precipitation and assumes an ice-free scenario before $32 \, \mathrm{ka}$ BP. Bedrock topography is adjusted for isostasy using the method of Oerlemans and van der Veen (1984).

The only global ice sheet models are the ICE-5G and ICE-6G, for the other ice sheet models we use the ICE-6G ice model outside the EISC. We include the build-up and deglaciation phase of the last glacial cycle. All ice sheet models are sampled in a grid with a spatial resolution corresponding to a $128$ degree Gaussian grid and the output of the model is truncated at degree $60$ and processed using the same filters used to process the GRACE data.

### 2.3 Model Performance Assessment

We assess the fit of the modelled and estimated gravity rates for different combinations of ice deglaciation history and rheology. As GRACE's resolution is of the same order of magnitude as the extension of the SBKIS we cannot resolve the differences in the shape of the ice sheet in the data. Thus we assess the model fit only by comparing the maximum modelled ($m_i$) and estimated ($e_i$) gravity rate in the central Barents Sea and Fennoscandia and normalize this difference using the observation

error ($\sigma_i$). In order to make the results as independent of the filter parameters as possible, we compute an average misfit using different filter configurations:

$$\chi^2 = \frac{1}{N} \sum_{i=1}^{N} \left( \frac{e_i - m_i}{\sigma_i} \right)^2, \tag{2}$$

where $N$ is the number of filter settings. The low-pass filter is varied between 200 and 300 km half width in 20 km intervals. Additionally, in the Barents Sea the high-pass filter is varied between the 500 and 700 km half width in 100 km intervals. We cannot formally define a confidence region as GRACE's observations processed with different filter configurations do not form a set uncorrelated observations. Instead, we define a subset of best fitting models as those that differ less than $2\sigma$ from the observations, indicating that, given the measurement noise, any of these models could be the best fitting model for a different realization of the observations. In the following we will refer to the lowest upper mantle viscosity of this set as lower bound

## 3 Results

### 3.1 GRACE GIA signal in Fennoscandia and the Barents Sea

We use the methods presented in Section 2 to obtain the gravity rates over Fennoscandia and the Barents Sea. A clear positive anomaly is evident both in Fennoscandia and the central Barents Sea where the main domes of the Scandinavian Ice Sheet and Svalbard-Barents-Kara Ice Sheet were presumably located (Figure 1). The melting of ice in Svalbard and the islands of the Russian Arctic Archipelago is also evident as a negative gravity trend. After removing the mass loss signal as explained in Section 2, we observe that most of the signal of Novaya Zemlya, Svalbard and Franz Josef Land is indeed removed (Figure 1). However, there is still a negative gravity rate left over Severnaya Zemlya, indicating that our ice loss changes might be underestimated for this island. We consider the remaining part of the signal to be entirely due to GIA and call it the estimated GIA signal. We do not observe a clear positive signal in the Kara Sea, which indicates that if it was glaciated during the LGM the amount of ice present there was much smaller than that located in the Barents Sea. This fact advocates against the larger ice sheets in Denton and Hughes (1981); Grosswald (1998); Grosswald and Hughes (2002) and further confirms the results of the DATED-1 (Hughes et al., 2016) and QUEEN projects (Svendsen et al., 2004).

We obtain the maximum gravity rate in the Barents Sea for different filter configurations using the OMCT and ECCO ocean models. Figure 3 shows the maximum gravity rates for a 600 km high-pass filter and different low-pass filter half widths. As expected, we observe that the maximum gravity signal reduces with increasing filter half width and so does the error. The gravity rates recovered using the ECCO ocean model are systematically higher that those obtained with the OMCT model. We show a breakdown of the error (Figure 4) for different low-pass filter half widths. We observe that the hydrology signal leaking into the Barents Sea is very small and the error budget is dominated by the uncertainty in present-day ice changes, the GRACE measurement error and the non-tidal ocean signal. Moreover, we observe that while the other error sources decrease with increasing filter half width the ocean error does not. This implies that it has a wavelength similar to that of the GIA signal we want to resolve.

## 3.2 Implications for viscosity and ice sheet chronology

We perform three experiments. In the first experiment we only study the effect of the ice history on the model misfit. We use the reference Earth model (see Table 2) and compare the fit of the predicted gravity rates for different ice deglaciation models with the GRACE derived gravity rate. In the second experiment, we change the Earth rheological parameters to obtain the subset of ice deglaciation histories and Earth rheological parameters that best fit the GRACE observations. Thirdly, we repeat the second experiment for Fennoscandia and compare the optimal solid Earth parameters for both regions to detect possible variations in rheological parameters.

Figure 3 compares the maximum present-day estimated gravity rates in the Barents Sea with those given by the different ice sheet models. It must be noted, that the maximum gravity rates produced by each ice history are not only related to the maximum ice volume attained during LGM, but also its geographical distribution and the onset of the deglaciation process. As an example, we find that while the T2 model has more ice in the Barents Sea than the T1 model, it results in lower gravity rates. This is because deglaciation starts earlier in the T2 model than in the T1 model when the sensitivity of the present gravity rates to mass changes is higher as shown in Figure 5. Similarly, the highest gravity rates are associated with the UiT ISM even though it has less ice than the ICE-5G and ICE-6G model. This is because the UiT ice sheet model has more ice in the central Barents Sea during the last phase of deglaciation. In fact, the model includes an ice bridge between Svalbard, Franz Josef Land and Novaya Zemlya with ice thickness as large as $2000\,\mathrm{m}$ at $14.5\,\mathrm{ka}$ BP which does not disappear until $12\,\mathrm{ka}$ BP. This is not present in either the ICE-6G or the ICE-5G models.

When we compare the modelled and estimated gravity rates we find that, for the reference Earth model, the T1, T2 and T3 ice sheet models are the closest to observations. The S04 ice sheet model performs worse; the model does not have enough ice in the region. This result is in accordance with Auriac et al. (2016) who found poor agreement between the S04 model and RSL curves. The more massive ICE-5G, ICE-6G and UiT models result in gravity rates that are too high. However, the discrepancy between these models and GRACE derived estimations is reduced if we use the ECCO ocean model instead of the OMCT. Furthermore, GRACE data can be reconciled with the UiT ISM if the maximum volume of mass in the model is reduced by around $1\,\mathrm{m}$ of equivalent sea level rise or if deglaciation started $1\,\mathrm{kyr}$ earlier.

Next, we study the effects of changing the solid Earth rheology in the Barents Sea. Figures 6 and 7 (left column) show the misfit of the different ice sheet models to the estimated maximum gravity rates in the Barents Sea for different rheology models. We see that there is a large subset of Earth rheological parameters for which the modelled gravity rate is within $2\sigma$ of GRACE's estimated gravity rate.

The T1, T2 and T3 ice sheet models present a good fit to the observations for a large subset of Earth models including the reference Earth model ($\nu = 5 \cdot 10^{20}\,\mathrm{Pa \cdot s}$, $h = 90\,\mathrm{km}$). For the less massive S04 model the $2\sigma$ interval extends from $\nu = 8 \cdot 10^{20}$ to $\nu = 1.6 \cdot 10^{21}\,\mathrm{Pa \cdot s}$. In contrast, for the more massive ice sheets (ICE-5G, ICE-6G and UiT ISM) the subset of Earth models which present a good fit to the Barents Sea observations is smaller and does not contain the reference Earth model. These models, however, fit the observations either for a less viscous upper mantle or for a thicker lithosphere when upper mantle viscosity is fixed. If a less viscous upper mantle viscosity is used the relaxation time of the solid Earth is decreased and the

sensitivity to mass changes that occurred during the LGM decreases (see Figure 5). On the other hand, a thicker lithosphere acts as a low-pass filter that smooths the gravity signal, reducing its maximum value.

Our results for the UiT ISM are consistent with those obtained by Patton et al. (2017) who inferred an upper mantle viscosity of $2 \cdot 10^{20}$ Pa$\cdot$s based on RSL data. The lower bound obtained with the other ice models is similar as models with a low viscosity have little sensitivity to mass changes during the early deglaciation phase, where differences between ice models are more manifest (Figure 2). Overall, for a lower mantle viscosity of $2.6 \cdot 10^{21}$ Pa$\cdot$s, we obtain a lower bound for the upper mantle viscosity of $3 \cdot 10^{20}$ Pa$\cdot$s, which agrees with the range of possible upper mantle viscosity found in Auriac et al. (2016) using RSL curves and GPS uplift measurements. We refrain from drawing conclusions on the preferred lithosphere thickness from the misfit plots because the lithosphere has a large influence on the shape of the gravity rate pattern which was not used as constraint here. A higher lower mantle viscosity can result in a lower upper mantle viscosity that still provides a good fit as shown in Steffen et al. (2010); Root et al. (2015b).

We follow the same procedure for Fennoscandia to obtain the subset of Earth rheological parameters and ice sheet deglaciation histories with an acceptable agreement with the GRACE observations (see right column of Figures 6 and 7). It must be noted that the values of the $\chi^2$ are higher for Fennoscandia than the Barents Sea and thus the subset of models within the $2\sigma$ contour is smaller. The reason is twofold: the observation error is smaller as compared with the Barents Sea, where uncertainty from mass changes in the glaciers of the surrounding islands and non-tidal ocean changes increase the error bars; and the GIA signal is higher in Fennoscandia than in the Barents Sea (see Figure 1). Nevertheless we can compare the best-fitting models for both regions.

We observe that, contrary to what we got for the Barents Sea, the combination of the ice sheet models ICE-5G and UiT with the reference lithospheric thickness and upper mantle viscosity have a good fit fit. (Figures 6 and 7). As already mentioned, the ICE-5G and ICE-6G models have been constrained using GIA observations, which are abundant in Fennoscandia. As we are using these models with an Earth rheology similar to its reference rheology it is not surprising that the ICE-5G model presents a good fit in this region, however the ICE-6G model performs better with a more viscous mantle due to its lower ice volume. The T1-3 models do not fit the estimated GIA signal with the reference Earth model and require a more viscous mantle. The early deglaciation and small LGM ice volume of the S04 model results in low gravity disturbance rates that do not fit the GRACE estimated gravity disturbance rates for the set of rheology parameters considered in this study. For Fennoscandia we find a lower bound for the upper mantle viscosity of $5 \cdot 10^{20}$ Pa$\cdot$s which is consistent with current estimates (Simon et al., 2018).

We can infer lateral rheology changes by comparing the optimal Earth rheological parameters obtained for both regions. For each ice deglaciation chronology, we compare the two $2\sigma$ intervals as well as the best fitting upper-mantle viscosity obtained for each lithospheric thickness. We observe that for the UiT, ICE-5G and ICE-6G model both the $2\sigma$ intervals as well as the best fitting models systematically prefer a less viscous upper upper mantle in the Barents Sea as compared with Fennoscandia. This is also the case for the T1,T2 and T3 models when the best fitting models are compared, although there is an overlap of models of high upper mantle viscosity and thick lithospheres with a good fit in both regions. This systematic difference is likely evidence of lateral variation in Earth rheology.

## 3.3 Lateral viscosity variation

To strengthen the conclusion of viscosity differences between the two regions, we derive viscosity estimates in an independent way, based on seismic velocity anomalies and experimentally derived flow laws. The absolute viscosity values obtained in this way contain large uncertainty, but the relative difference resulting from the seismic models should represent real change in temperature or composition. Therefore we focus on the ratio between the viscosities beneath Fennoscandia and the Barents Sea and check whether it agrees with the outcome of the GIA model misfit.

To take uncertainty in seismic velocity anomalies into account we use two global seismic tomography models: S40RTS Ritsema et al. (2011) and Schaeffer and Lebedev (2013) (labeled SL) which has higher spatial resolution but reduced sensitivity with depth. For both, the reference model is adjusted to account for a jump in the seismic velocity anomaly in PREM (Dziewonski and Anderson, 1981) and AK135 (Kennett et al., 1995) below 200 km. Shear wave velocities are converted to temperature using relations from geochemistry (Goes et al., 2000; Cammarano et al., 2003) for primitive mantle composition and accounting for anelasticity (anelastic correction model Q4 from Cammarano et al. (2003)). Differences in composition between the Barents Sea and Scandinavia could play a role, but is unlikely to reverse the temperature contrast, due to the first order effect of temperature on seismic velocities in the upper mantle (Goes et al., 2000). To compute viscosity we follow the procedure in Wal et al. (2013) and insert temperatures in the olivine flow laws of Hirth and Kohlstedt (2013). The flow laws for diffusion and dislocation are added, which means the viscosity depends on grain-size and stress. Stress is taken from a 3D GIA model which uses the ICE-5G ice load. Background stresses due to mantle convection are neglected as recent work suggest little interaction between GIA and mantle convection (Huang, 2018). Grain size is chosen to be 4 mm or 10 mm. 4 mm gave best overall fit to GIA data in and 10 mm grain size resulted in the best fit with the observed maximum uplift rate (Wal et al., 2013).

To be able to compare against viscosity for the upper mantle in the previous section we use viscosity averaged between 225 and 325 km. This depth is a trade-off; shallower layers have lower temperature and small viscous deformation during the glacial cycle, while for deeper layers the seismic models are less accurate. The depth range is also close to the depth to which the gravity rate in Fennoscandia is most sensitive, see the sensitivity kernels in van der Wal et al. (2011). The viscosity maps are plotted in figure 8. In principle all viscosity values around the ice load play a role in the GIA process, but the highest sensitivity is to values directly underneath the ice load (Paulson et al., 2005; Wu, 2006). We compute the average of viscosities for the locations where LGM ice heights are above 1500 m which covers most of the land mass of Scandinavia, and most of the Barents Sea (see dashed brown contour). Viscosity is computed separately for the region below 71° latitude for Fennoscandia and above for the Barents Sea. We find that the average viscosity below Fennoscandia is a factor of 2.3 to 2.4 times higher than that in the Barents Sea. This agrees well with the change in best fit upper mantle viscosity that can be seen in the misfit figures 6 and 7. There could still be an effect of 3D structure that is not captured by modelling both regions with 1D models, such as lateral variations within Fennoscandia (Steffen et al., 2014) or the influence of viscosity from outside each region.

## 4 Conclusions and Discussion

In this study, we analyse GRACE data in the Barents Sea to constrain the Earth rheology in the region. We compare the fit of different GIA models in Fennoscandia with that for the Barents Sea to find if there is a difference in viscosity between the two regions. We investigate several deglaciation chronologies of the SBKIS, some of which are not explicitly tied to a viscosity model. We use GRACE data for the period $2003 - 2015$ and process it to reveal the GIA signal. The ice loss signal from the Svalbard and the Russian Arctic Archipelago is removed using mass change values obtained from GRACE using the mascon method. We observe a positive gravity anomaly in the Barents Sea but no significant anomaly in the Kara Sea, which shows that the ice cover at LGM was considerably thinner there than in the Barents Sea, in agreement with recent studies.

The Barents Sea GIA signal is in a region now covered by sea; therefore, the gravity trends might be affected by non-tidal oceanic mass changes. We correct GRACE gravity rates in the Barents Sea using either of two ocean models, the OMCT and ECCO ocean model, and find higher gravity rates using the ECCO model. The difference in the ocean signal according to the two models is large in the Barents Sea. This uncertainty has not been considered in previous studies of the GIA signal in the region (e.g., Root et al., 2015a; Simon et al., 2018; Kachuck and Cathles, 2018) and thus the errors bars in those studies were underestimated. This result has also implications for GRACE studies of non-oceanic mass changes, such as post seismic deformations, in ocean areas (e.g., Han and Simons, 2008; Wang et al., 2012) which possibly have higher uncertainty than previously thought due to errors in the ocean model.

We compare the GRACE derived gravity rates with modelled ones to infer geophysical constraints for the Earth rheology and ice sheet chronology in the Barents Sea region. For a three-layer average of the VM2 viscosity profile (Peltier, 2004) we find, as Root et al. (2015a), that thick ice sheet models (ICE-5G, ICE-6G and UiT) do not fit GRACE observations, while the less massive ice models (T1,T2 and T3) do. Upper mantle viscosity and lithospheric thickness was varied for each ice sheet chronology between $0.1 \cdot 10^{21} - 32 \cdot 10^{21}$ Pa $\cdot$ s and $40 - 180$ km. We find that the ICE-5G, ICE-6G and UiT ice sheet models can be reconciled with GRACE observations provided the upper mantle viscosity is lower or the lithosphere thicker than in the VM2 model. The same conclusion is reached in Auriac et al. (2016) using GPS uplift measurements and RSL curves instead of gravity data.

The interplay between ice deglaciation chronology and Earth rheology makes it difficult to constrain the ice deglaciation chronology in the Barents Sea (Kachuck and Cathles, 2018). Root et al. (2015a) used GRACE data to conclude that the SBKIS had less ice than previously thought ($5 - 6.3$ m of equivalent sea level versus $8.3$ m). To do so, they used ICE-5G and ICE-6G and showed that they do not obtain the estimated gravity rate when these ice models are combined with their corresponding Earth rheology model. However, here we use the UiT ISM which does not come with an *a priori* Earth rheology model and which contains around $7.5$ m of equivalent sea level rise and show that it can fit GRACE observations provided the upper mantle viscosity is around $3 \: 10^{20}$ Pa $\cdot$ s if the lithosphere is thinner than $130$ km. However, we are able to place a constraint on upper mantle viscosity. From the misfit of all investigated ice chronologies and using a lower mantle viscosity of $2.6 \cdot 10^{21}$ Pa $\cdot$ s, we find that best fitting models have an upper mantle viscosity equal to or higher than $3 \cdot 10^{20}$ Pa $\cdot$ s agrees with previous constraints derived from RSL and GPS uplift observations Auriac et al. (2016).

We also study the misfit of GRACE observations to the GIA models in Fennoscandia. For a $2.6 \cdot 10^{21}$ Pa·s lower mantle viscosity, best fitting models have an upper mantle viscosity equal to or higher than $5 \cdot 10^{20}$ Pa·s, which is consistent with current estimates. Given all the ice sheet deglaciation chronologies we find that the lower bound for the upper mantle viscosity is a factor of two smaller in the Barents Sea (or, alternatively, the lithosphere thickness should be increased there). Unless all

the tested ice deglaciation chronologies are biased in the same direction, this result is evidence of lateral changes in viscosity in between the two regions.

To strengthen the finding of viscosity difference between the two regions, we compare our results with viscosity derived from global velocity anomalies and flow laws for mantle material and find that the average viscosity in the Barents Sea is a factor of 2.4 lower than in Fennoscandia. This agrees very well with the results derived from the misfit of GIA models to

GRACE data, and strengthens the conclusion that there is a small but significant difference in average upper mantle viscosity between the two regions. This findings have implications for ice sheet models inverted with just one viscosity profile (e.g., ICE-5G, ICE-6G) and advocates in favour of including lateral Earth rheological parameters in GIA models. The constraints on viscosity variations can be also used to calibrate other geodynamic models of the regions.

*Code and data availability.* Gravity rates for the different ice sheet models and Earth rheology models as well as GRACE maximum distur-

bance rates for Fennoscandia and the Barents Sea are provided at http://doi.org/10.4121/uuid:424126e6-b5d3-4ac9-b5cd-f495c8ad6939. The GIA code used for the simulations is available upon request from VRB.

*Author contributions.* All authors contributed to the discussion and commented on the manuscript. M.R.N and W. v.d.W led the writing of the article. V.R.B contributed with her GIA code. M.R.N analysed GRACE data and ran the GIA simulations. W.v.d.W. provided the 3D viscosity maps. All authors contributed to the interpretation of the results.

*Competing interests.* The authors of this manuscript declare that they do not have any conflict of interest.

*Acknowledgements.* The authors would like to thank L. Tarasov, H.Patton and M. Siegert for making their ice sheet models available for this study (T1,T2 and T3; UiT ISM and S05 model). The authors also thank E.J.O. Schrama for providing mass loss changes in the islands of the Arctic Archipelago for this work, and W. Stolk for his contribution to the viscosity maps. The authors also thank I. Fenty for his assistance and advice on the ECCO ocean model products. The first author would like to thank Fundacio la Caixa for the financial support he received

while conducting this research.

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

**Table 1.** Ice loss changes in Svalbard the Islands of the Russian Arctic Archipelago between 2003 and 2015 in $\mathrm{Gt/yr}$ obtained for different GIA models. The ICE-5G model and two runs of the GSM with maximum (GLAC2) and minimum (GLAC1) ice sheet extents that comply with RSL and GPS observations combined with the VM5 Earth rheological model or a model with stronger mantle, labelled M2, with $\mu_{UM} = 1.6 \cdot 10^{21}$ Pa$\cdot$s and $\mu_{LM} = 5.12 \cdot 10^{22}$ Pa$\cdot$s. Additionally, the W12 ice model with $\mu_{UM} = 1 \cdot 10^{21}$ Pa$\cdot$s and $\mu_{LM} = 1 \cdot 10^{22}$ (M3) is also used. The last row indicates the average value and uncertainty due to GRACE measurement error and uncertainty in the GIA model

| Ice Model | Rheology | Novaya Zemlya | Svalbard | Franz J.Land | Servernya Zemlya |
|---|---|---|---|---|---|
| GLAC1 | M2 | $4.71 \pm 0.42$ | $5.05 \pm 0.49$ | $1.12 \pm 0.19$ | $0.76 \pm 0.10$ |
| GLAC1 | VM5a | $4.94 \pm 0.42$ | $4.96 \pm 0.49$ | $1.10 \pm 0.19$ | $0.63 \pm 0.10$ |
| GLAC2 | M2 | $4.60 \pm 0.42$ | $4.90 \pm 0.49$ | $0.93 \pm 0.19$ | $0.92 \pm 0.10$ |
| GLAC2 | VM5a | $4.57 \pm 0.42$ | $4.85 \pm 0.49$ | $0.85 \pm 0.19$ | $0.67 \pm 0.10$ |
| ICE-5G | M2 | $5.87 \pm 0.42$ | $5.77 \pm 0.49$ | $1.68 \pm 0.19$ | $0.80 \pm 0.10$ |
| ICE-5G | VM5a | $4.54 \pm 0.42$ | $5.16 \pm 0.49$ | $1.03 \pm 0.19$ | $0.70 \pm 0.10$ |
| W12 | M3 | $6.13 \pm 0.42$ | $5.34 \pm 0.49$ | $1.64 \pm 0.19$ | $0.46 \pm 0.10$ |
| - | - | $5.15 \pm 0.58$ | $5.05 \pm 0.79$ | $1.19 \pm 0.38$ | $0.70 \pm 0.18$ |

**Table 2.** Solid Earth rheological parameters for this study: lithosphere thickness ($h_l$), upper mantle viscosity $\nu_{UM}$ and lower mantle viscosity $\nu_{LM}$

| Parameter | Reference Model | Range |
|---|---|---|
| $h_l$ (km) | 90 | $40 - 180$ |
| $\nu_{UM}$ ($10^{21}$ Pa$\cdot$s) | 0.5 | $0.1 - 1.6$ |
| $\nu_{LM}$ ($10^{21}$ Pa$\cdot$s) | 2.6 | 2.6 |

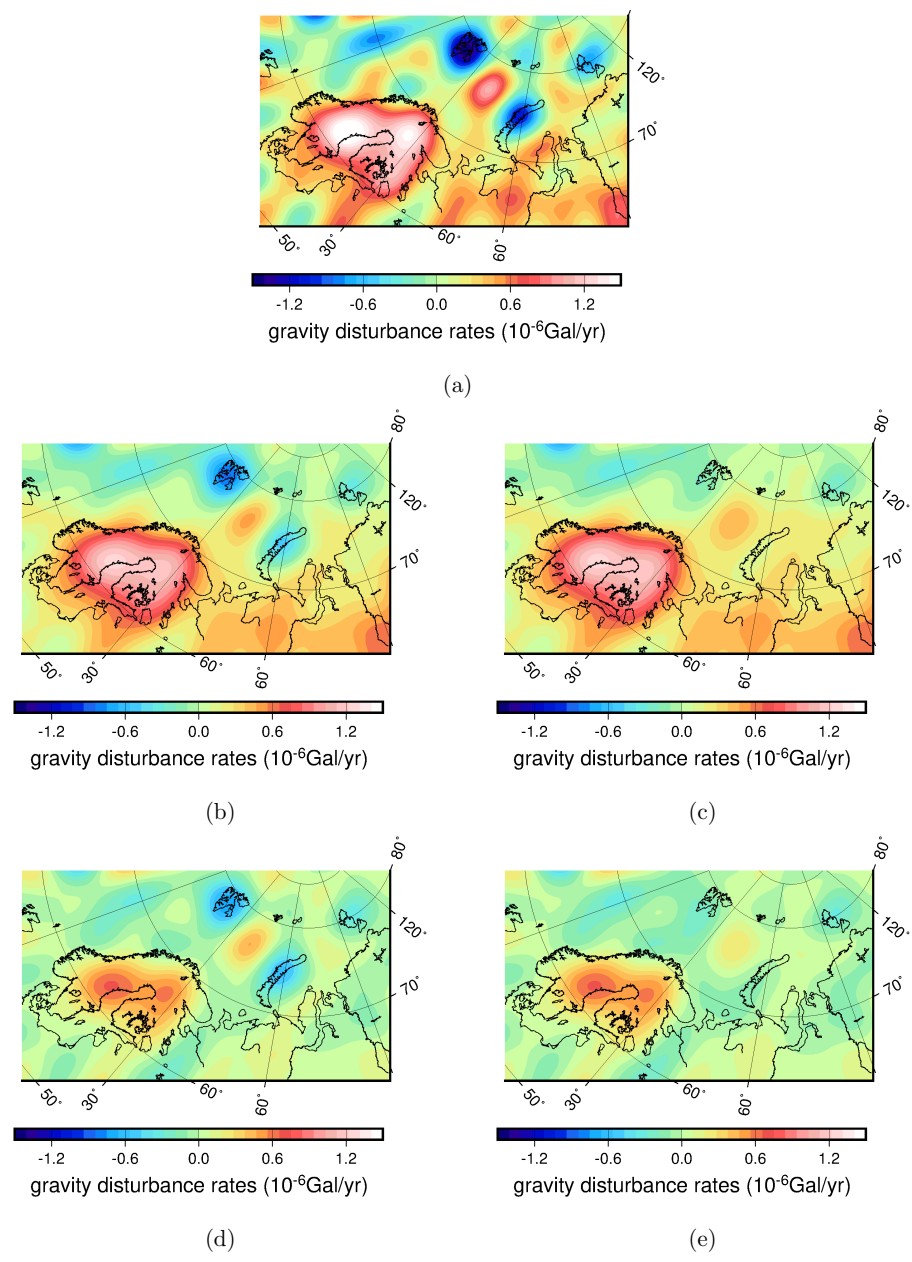

**Figure 1.** Gravity signal in Fennoscandia and the Barents Sea for the period 2003-2015. (a) shows the gravity disturbance trends for the unprocessed GRACE data. (b) and (c) show the gravity disturbance rate filtered with a 200 km low-pass filter while in (d) and (e) the data is additionally filtered with a 600 km high-pass filter to remove long wavelength signals. The mass loss signal of the Arctic Archipelago islands has been removed in (c) and (e).

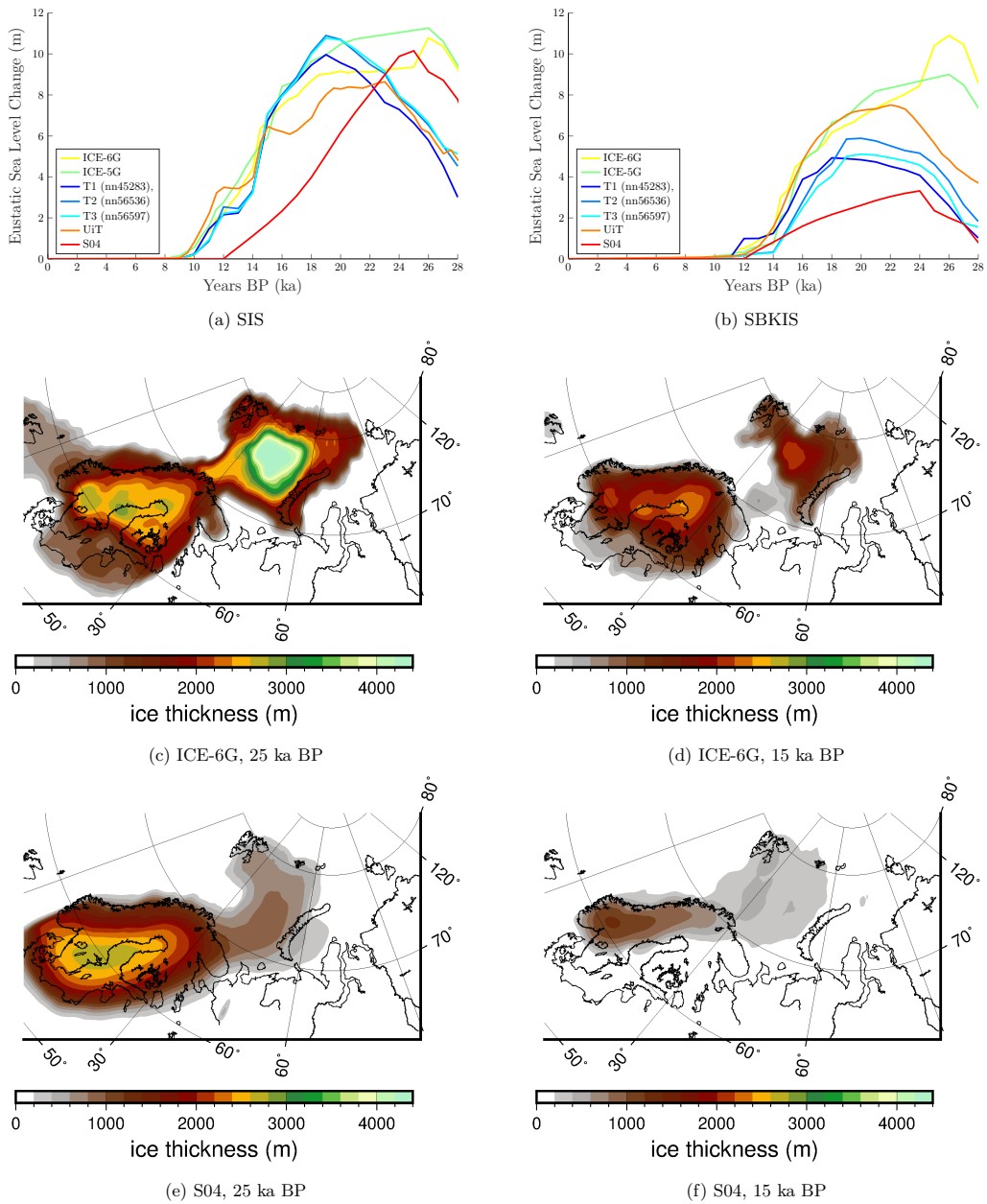

**Figure 2.** Volume of ice present in the SIS (a) and SBKIS (b) during the last glacial period given in equivalent eustatic sea level rise for different ice sheet reconstructions. Six different deglaciation chronologies are shown: the GIA-constrained models ICE-5G and ICE-6G (Peltier, 2004; Peltier et al., 2015; Argus et al., 2014); three models obtained using the Glacial System Model (GSM) (Tarasov et al., 2012), the T1, T2 and T3 chronologies; the University of Tromsø Ice Sheet Model (UiT) (Patton et al., 2017); and the S04 ice sheet model (Siegert and Dowdeswell, 2004). The divide between both ice sheets is taken to be the 70° parallel. Ice extent and thickness are shown for the ICE-6G (c,d) and S04 (e,f) ice models for two different epochs.

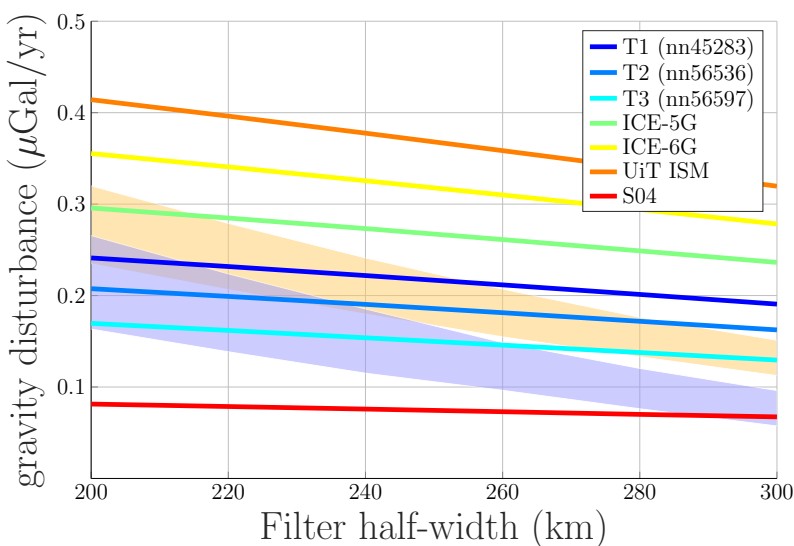

**Figure 3.** Maximum gravity rate in $\mu$Gal/yr recovered in the central Barents Sea using GRACE, after removing the ocean signal with the OMCT ocean model (blue) or ECCO ocean model (orange) for different low-pass filter half widths and a 600 km half width high-pass filter. The GIA signal for different ice deglaciation histories with the reference Earth model is also shown.

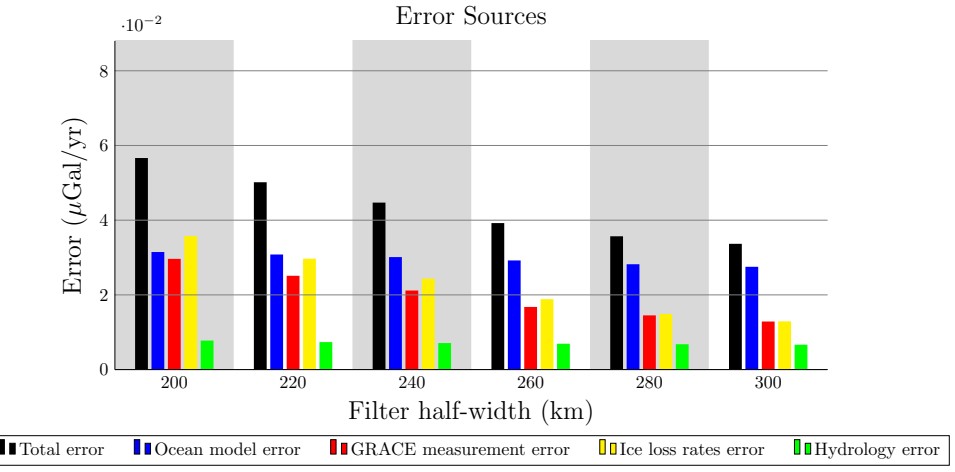

**Figure 4.** Error in $\mu$Gal/yr in the maximum gravity rate in the central Barents Sea from different sources. The magnitude of the error is given for different low-pass filter half widths and a high-pass filter half width of 600 km.

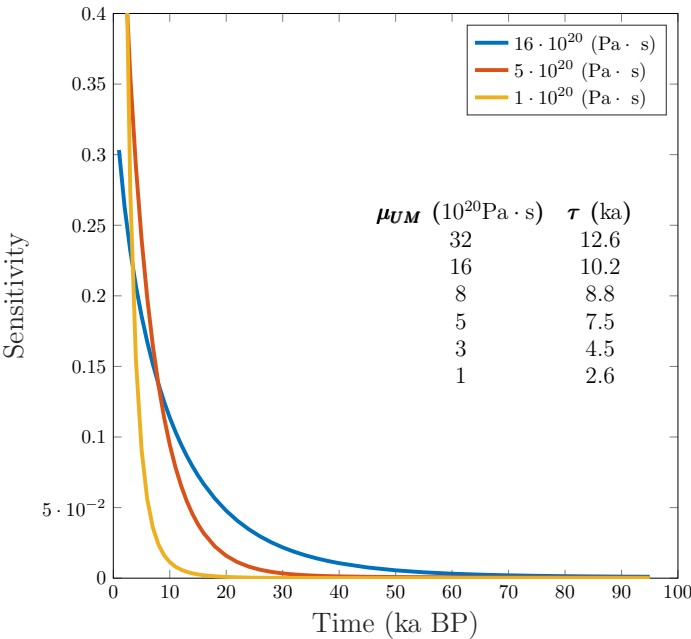

**Figure 5.** Present gravity disturbance rate induced by a uniform mass change in the Barents Sea at a given epoch for three different upper mantle viscosity. The results have been normalized using the maximum gravity disturbance rate obtained with $\mu_{UM} = 1 \cdot 10^{20}$ Pa·s. Inset: relaxation times for different upper mantle viscosity.

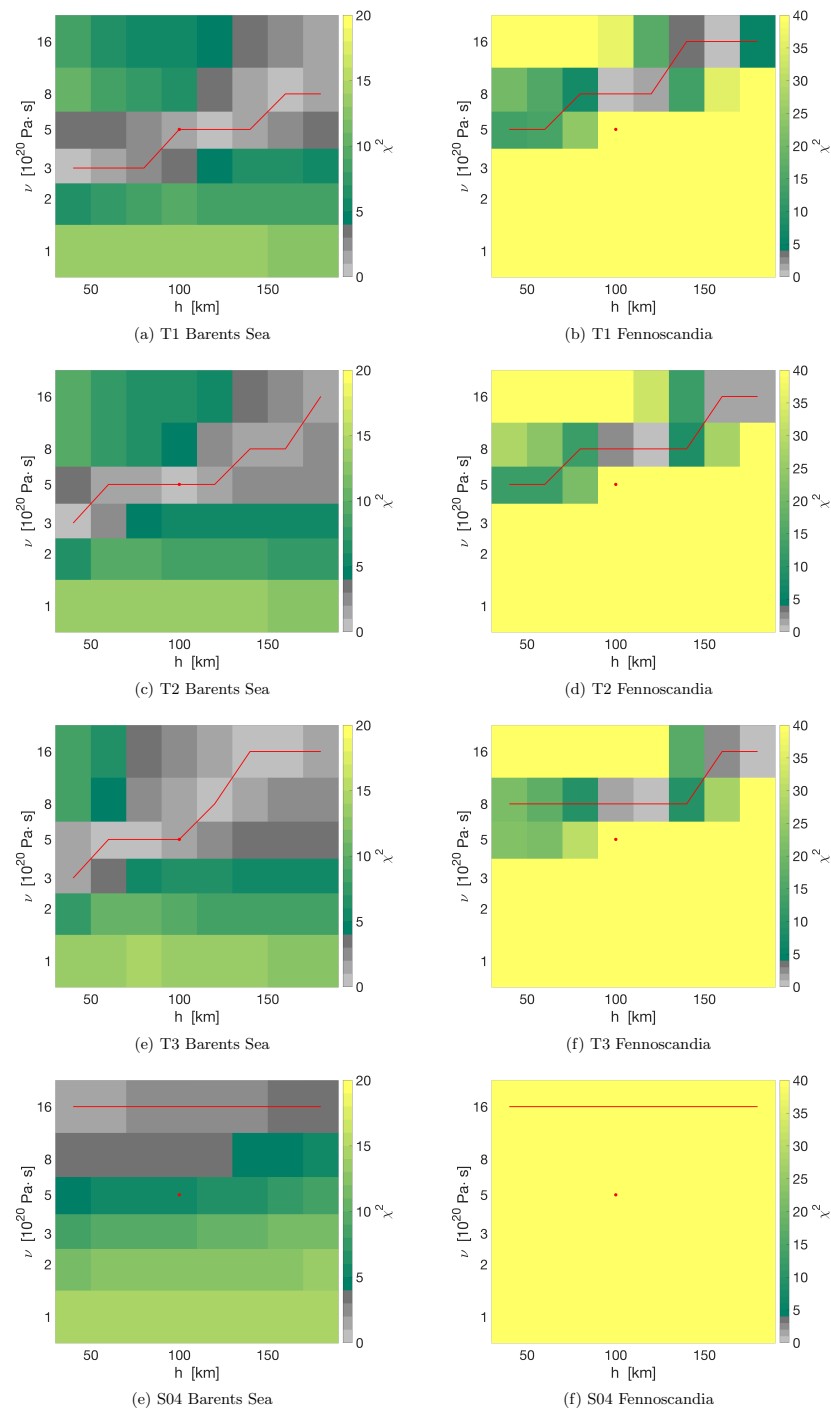

**Figure 6.** Misfit of the T1,T2,T3 and S04 ice deglaciation chronologies to GRACE observations for different values of upper mantle viscosity ($\nu$) and lithospheric thickness ($h$) in the Barents Sea (left column) and Fennoscandia (right column). The fit is given in terms of the $\Delta\chi^2$. The circle indicates the reference model and the red line shows the best fitting model for each lithospheric thickness

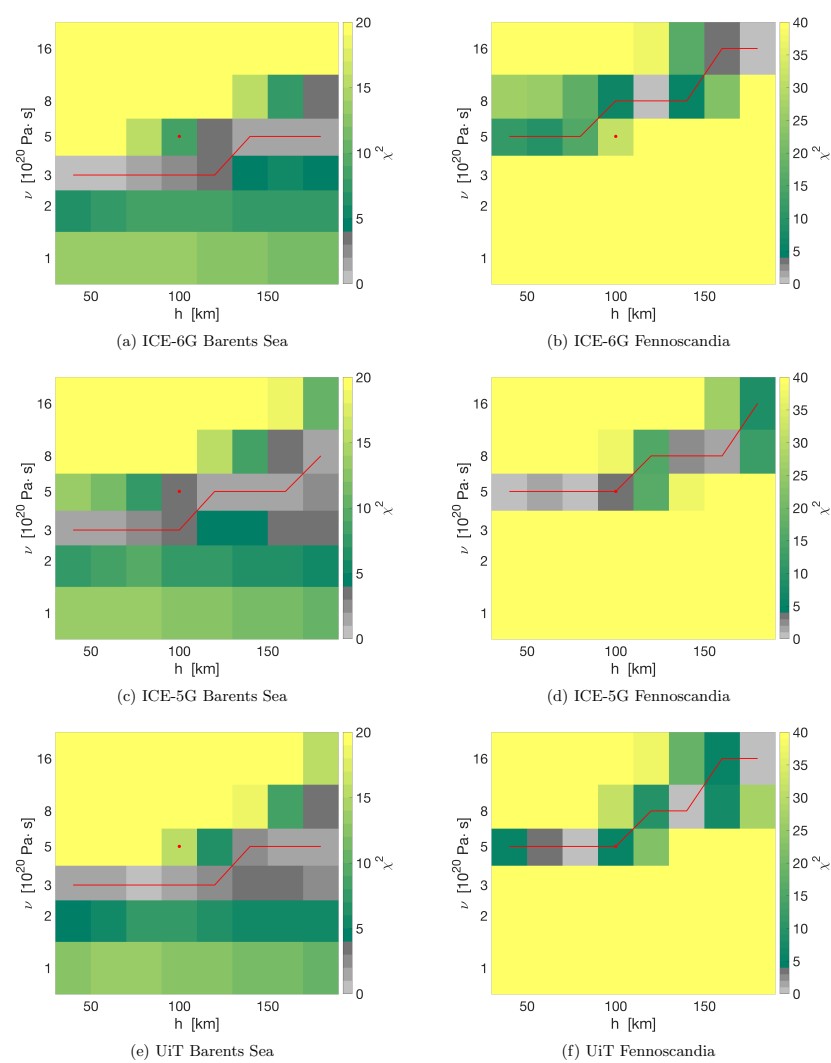

**Figure 7.** Same as Figure 6 but for the ICE-6G, ICE-5G and UiT ice sheet models.

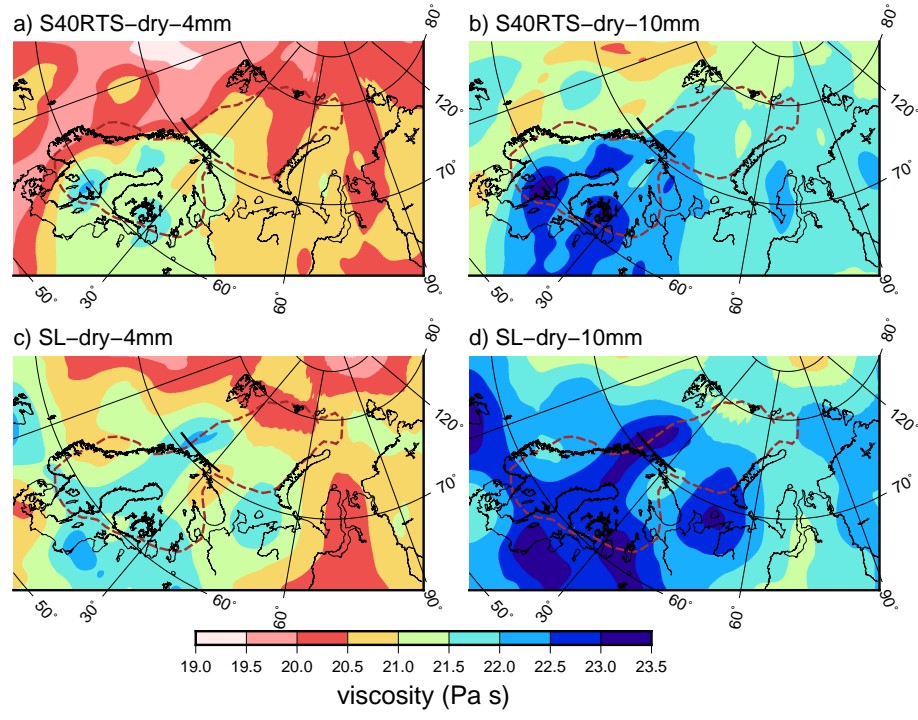

**Figure 8.** Viscosity between 225 and 325 km depth derived from seismic models S40RTS (Ritsema et al., 2011) (a and b), and Schaeffer and Lebedev (2013) (c and d), and for different flow law parameters: 4 mm grain size (a and c) and 10 mm grain size (b and d). The brown line denotes the 1500 m ice height contour at LGM in the ICE-5G model; the black line denotes 71° latitude which separates the areas used for computing the viscosity for Fennoscandia and the Barents Sea.