# Peer review of "GRACE constraints on Earth rheology of the Barents Sea and Fennoscandia"

_Solid Earth, 2019_

## Referee Comment (RC1) · Anonymous Referee #1 · 11 Jul 2019

"GRACE constraints. . ." by Rovira-Novarro, et al., investigates the constraints provided by GRACE measurements on upper mantle Earth rheology in the Barents Sea. Using a sample of deglaciation chronologies and a grid-search of upper mantle viscosity and lithospheric thickness, they identify a lower bound on viscosity and evidence of lateral heterogeneity moving from the Barents Sea to Fennoscandia. The investigation is a follow-on to the paper Root, et al. 2015a, contributing the parameter search in earth response modeling, additional ocean modeling, and a discussion of lateral heterogeneities. It is an important step in quantifying the uncertainties in rheology associated with matching the changing gravity field in the Barents Sea. However, I am concerned that the result does not accurately represent the state of knowledge. As I describe below, the result seems to rely on aggressive processing and a limited, pos-

sibly biased, range of ice models. With these issues addressed, I would look forward to seeing this article in press.

General Comments

My primary concern is about the substantial signal lost through the filtering and processing of the measurements and low resolution of the modeling. The limitation of maximum order number 60, which yields a minimum resolution of approximately 300 km, or one quarter of the linear extent of the Barents Sea. It is also cuts off a significant potion of the power of the authors' bandpass filter ranges. Thus the shape of the bandpass filter dominates the shape of the processed and modeled measurements.

This filtering occurs after a series of processing steps to extract the LGM signal. The GRACE measurements are processed one way to estimate the current mass loss off the archipelagos, another to estimate the ocean signal, and a final way to estimate the response to LGM deglaciation in the Barents. So, while I really appreciate the attempt to quantify all of the sources of error, the assumption that they are uncorrelated (page 5, line 12) requires further explanation. I would similarly like elaboration of the effect the GIA model chosen has on the estimate of mass loss (page 4, line 33).

In light of this concern, I would ask the authors to: 1) further quantify the effects of their processing technique for this area. In particular, by adding more discussion of the technique for idealized measurements in the context of the Barents; and 2) consider acknowledging the processed nature of these results by referring to them as "estimated gravity rates" rather than "observed gravity rates." I feel this is particularly important when the authors substitute the phrase "observed gravity rate" for the estimated maximum gravity rate (e.g., page 6 - line 31).

The argument additionally suffers from another small, but troubling, circularity. The ensemble of ice models was chosen to represent two classes: empirical ice sheets developed using GIA observables and ice sheets developed from independent, process based models. However, all of the models are actually calibrated, in some way or

another, to GIA observables with an implicit dependence on the assumed viscosity structure. For instance, the Tarasov samples are drawn from a distribution trained on GIA observables using the Peltier VM5a rheology. If the authors could comment on this bias and how that might account for the reference model being very near the best fit valley in all figures but the Siegert and Dowdeswell 2004 model, which is the only one to prefer an anomalously high viscosity, most likely because of it's earlier ice-free time.

Specific comments page-line 4-33: "However, the GIA" It is not obvious that this should be so. 8-19: citing the \chi^2 might make this point clearer. It is hard to tell that S04 is significantly worse than, say UiT, from figure 3 10-32 might include "explicitly" in "not explicitly tied to a viscosity model" Figure 5 and Figure 6 - Could you note with a symbol the reference model and the best fit model in each of these plots?

Technical corrections page-line 1-2: in-> to "insight to the" 1-4: Split sentence 1-5 remove "a" in "a GIA models" 1-6 "is not negligible" and "should be taken into account" are redundant 1-16 Inconsistent use of "gravity disturbance rate" and "gravity rate" 3-7 missing word in "while best fitting models uplift rate measurements" 3-22 missing "and" in "GIA, and (post-) seismic" 5-4 GAB undefined 5-9 "respectively" has no antecedent. Consider "both the OMCT and ECCO ocean models" 5-15 "while when" is difficult to parse 5-22 missing word in "This still allows" 5-23 correct citation parenthesis 5-25 missing "the" in "that of the unknown" 5-26 remove nested parentheses 5-31 missing "the" in "the Earth's rheology" 8-12 I believe deglaciation starts earlier in T2 than in T1, unless I am much mistaken. 8-31,8-32,9-30 "lower upper mantle viscosity" is pretty cumbersome to read. Consider something like "less viscous upper mantle" 9-9 repeated word "which that" 9-12 large->high "high upper mantle viscosity" 9-16 typo "form"

Figure 3 and Figure 4 - inconsistent x-axis label

---

## Referee Comment (RC2) · Anonymous Referee #2 · 28 Aug 2019

The manuscript by Rovira-Navarro et al., "GRACE constraints on Earth rheology of the Barents Sea and Fennoscandia" analyzes GRACE gravity data in order to constrain glacial isostatic adjustment (GIA) models in the two regions. They employ a number of different reconstructions of the Weichselian ice sheet and explore the effects of varying upper mantle viscosities and elastic thickness. The manuscript concludes a lower bound on the upper mantle viscosity of the the Barents Sea region and that that viscosity is a factor of 2 lower than the viscosity of the Fennoscandian upper mantle.

I find the manuscript well written and easy to follow. The application of GRACE data to an oceanic region like the Barents is important in order to gain a better understanding of the ongoing geodynamics and the extended uncertainty analysis in the manuscript seems like an appropriate way to go. I do find the manuscript a little brief on some

points which I detail below.

1. My main concern is the lack of discussion of uncertainties in the resulting viscosities. The conclusion that Fennoscandian upper mantle viscosity is a factor of 2 higher than that in the Barents Sea is given with very little discussion on uncertainties:

a) Most studies would state resulting viscosities and elastic thickness as an interval determined one way or another from the statistics of the inversion process. A differently normalized $\chi^2$ range or a variance reduction, for example. On page 8, line 27, a 2 sigma interval is mentioned but not further referred to. The very different $\chi^2$ distributions for Barents Sea and Fennoscandia in Figures 5 and 6 make it difficult to asses which parts of the model space is appropriate to compare to one another. In addition, the color scale in the Figures does not enhance the well fitting regions very well, I suggest a scale with a better visible range.

b) At least for the T1-T3 and S04 ice histories, even though the well fitting viscosity range starts at lower viscosities for the Barents Sea than for Fennoscandia, there is significant overlap at higher viscosity in Figure 5. This is less pronounced for ICE-XG and UiT in Figure 6, but is there at thicker elastic thickness. A more well defined range of which models are considered good fits would ease the comparison.

c) The lower bounds on viscosity is very similar for all ice models in the two regions. That is a little odd. Is there some bias somewhere? Such as they having similar Earth models during construction?

d) Elastic thickness is discussed very briefly in the manuscript. The clear correlations in Figures 5 and 6 between viscosity and elastic thickness should be discussed further. This is different to the results in e.g. Steffen et al. (2010), Root et al. (2015a,2015b). How much are the results for a thicker elastic layer affected by the GRACE filtering process? In addition, there are surely estimates from seismology of the (seismic) thickness of the lithosphere in the Barents and in Fennoscandia. These could perhaps also be used for comparison purposes, even though the measure a slightly different property.

e) When concluding the factor of 2 viscosity difference between Barents Sea and Fennoscandia you should specify at which elastic thickness the comparison is made. If you use different elastic thickness for the different regions that should be explicit. Similarly for the seismic estimates of viscosity difference. These are at the same depth for the Barents Sea and Fennoscandia but to make a fair comparison it would be interesting with estimates of the seismic lithosphere thickness. How much does it matter for the comparison if there are differences in temperature and/or composition in the two regions?

f) You should compare your inferred viscosity differences to other GIA studies of the Barents Sea area and Fennoscandia. The large number of varying results for Fennoscandia indicate that such a comparison is non-trivial. For the Barents Sea, Root et al. (2015a) indicates 4x10ˆ20 Pa s for the Barents and Auriac et al. (2016) has a very wide range of 2 - 20 x 10ˆ20 Pa s.

2. The four error estimates for the GRACE processing seem very appropriate, but I wonder:

a) Is spherical harmonic degree 60 really enough for this study? And the filtering out of smaller wave-lengths seems to retain only very large scale features, on the order of the whole basin?

b) The statement on page 5, line 12, about the independence of the estimates. It seems to me a little strange that the hydrological signal would be uncorrelated with the ice loss signal? Ice mass loss usually means melting, which surely influences the hydrology, both in time and magnitude. Is this not an issue?

c) The estimate of ice mass loss from GRACE data does indeed seem circular, and a little difficult. A GIA model using GRACE data is used to estimate uncertainties in the GRACE data for GIA applications? On page 4, lines 21-22 the authors state that the current ice mass changes "...partly mask the GIA signal...", but on lines 33-34 that "... the GIA model used to obtain the mass changes has a small effect in

the recovered gravity rate trend...". This seems contradictory to me and need more detailed explanation. Do you use different GRACE filters here to capture the spatially smaller current deglaciations? Also, how are the error bars estimated from the range of ice and Earth models? Do you have a range of reasonable chiˆ2 or something for this error estimate?

Detailed comments: - Including GPS data from Svalbard and northern Norway would have been helpful to constrain the models. Why was this not done? (See Auriac et al. (2016)) - It would be good to have a little bit more information on the ice reconstructions, especially with regard to the used Earth model physics for the non-GIA derived ice models. Do they have appropriate viscoelastic earths, or just simple hydrostatic adjustment, or...? Also, which time period do you use in the models? Just the deglaciation phase? If so, how are the ice sheets ramped up to the last glacial maximum? - Use regular non-italic font for units. Even in latex "Pa.s" can be made roman in math mode (if that is the problem)

Page Line Comment 1 3 You write insight into sub-surface structure. It is not really structure but rather rheology. 2 30 Same as above 1 4 Either spell out GRACE, or add "gravity" for clarity. 1 8 I would remove "deglaciation" here and describe the used time period in the paper. 1 16 Just to be clear, spell out GRACE or add gravity here the first time it is mentioned. 2 7 Here you could include the dynamic ice sheet model by Näslund et al. (2005): Näslund, J.-O., Jansson, P., Fastook, J. L., Johnson, J., and Andersson, L.: Detailed spatially distributed geothermal heat-flow data for modeling of basal temperatures and meltwater production beneath the Fennoscandian ice sheet, Ann. Glaciol., edited by: MacAyeal, D. R., International Glaciological Society, 40, 95–101, doi:10.3189/172756405781813582, 2005. 3 7-8 "... best fitting models uplift rate measurements..." is difficult to understand. 3 29 "... we use ..." the software? There is an object missing in the sentence. 4 6 Define "gravity disturbance rate" as opposed to "gravity anomaly rate". 5 3 Reference to the ECCO model. 5 4 What are the GAB products? 5 8 No italics. 5 17 Ocean bottom pressure changes in the Baltic can be

neglected? Are they so much smaller than in the Barents, or just relatively smaller? 6 28-29 This sentence need reformulation. 7 1 Are you using central Fennoscandia? If so, where is this? 7 9 In Figure 1 it is the gravity signal after processing, not necessarily the GIA signal. 9 5 "A second set..." Which is the first? 9 9 ".. gravity rate which that is larger than..." Fix this. 9 20 The authors should point out that 3D effects are indeed significant, e.g. Whitehouse et al. (2006), Steffen et al. (2006). 10 2 Which conclusion? 10 9 "...the reference model... a jump below 200 km" Please clarify which reference model and what the jump is, or refers to. 10 14 Stress for the flow law is taken from the GIA model. How accurate is this? Neglecting tectonics, topography, sediment loads etc surely distorts the "correct" stress state. How important is this? 10 23 Why did you choose the 1500 m contour? 10 31 You should define "significant", or rather add uncertainties. 12 1 "... the GRACE misfit"? The GRACE GIA models?

---

## Author Comment (AC1) · 3 Oct 2019

We thank the reviewer for the comments provided, we think they helped improving the quality of the manuscript and clarify some relevant points. Below, we address the reviewers' comments (in blue). A manuscript with the changes done to the manuscript shown in blue is also provided. References to pages and lines (page,L line) refer to the new version of manuscript, lines and pages mentioned in the reviewer's comments correspond to the first version of the manuscript

**Reviewer 1: Major comments**

Reviewer 1 express his concerns about some of the processing techniques used in this study. Below we address these comments:

My primary concern is about the substantial signal lost through the filtering and processing of the measurements and low resolution of the modeling. The limitation of maximum order number 60, which yields a minimum resolution of approximately 300 km, or one quarter of the linear extent of the Barents Sea. It is also cuts off a significant potion of the power of the authors' bandpass filter ranges. Thus the shape of the bandpass filter dominates the shape of the processed and modeled measurements.

The cut-off degree and filters used in GRACE data are equally applied to the simulated GIA signal (7, L22). This way the comparison of GRACE and simulated gravity rates is consistent

The effect of cutting the GIA signal at degree 60 is illustrated in Figure 1 and 2. Figure 1 shows the GIA signal obtained with the ICE-5G model cut at different degrees, Figure 2 gives the maximum gravity disturbance rate obtained in the Barents Sea for different cut-off degrees. Both plots evidence that the GIA signal does not have a high content of high degree harmonics and therefore little signal is lost.

The filters used to process the data are carefully chosen following the work of Root et al. 2015 (see supplementary data). A low pass filter is needed to filter out small wavelength noise. We vary the filter halfwidth between 200km and 300km. Below 200km noise becomes dominant and above 300km the positive signal located in the Barents Sea is very small (see Figure 3). The half-width of the high-pass filter is chosen using Figure 1 from Root et al. 2015 supplementary material. They use a synthetic GIA signal to show the effect of using a high-pass filter. From that figure it is concluded that a high pass filter between 500 and 700 km keeps the GIA signal while removing other long-wave signals.

This filtering occurs after a series of processing steps to extract the LGM signal. The GRACE measurements are processed one way to estimate the current mass loss off the archipelagos, another to estimate the ocean signal, and a final way to estimate the response to LGM deglaciation in the Barents. So, while I really appreciate the attempt
to quantify all of the sources of error, the assumption that they are uncorrelated (page 5, line 12) requires further explanation. I would similarly like elaboration of the effect the GIA model chosen has on the estimate of mass loss (page 4, line 33).

It is true that GRACE is used to (1) recover the LGM signal, (2) obtain the mass loss changes in the islands of the Arctic archipelago and (3) in a smaller degree in the ECCO model which makes it possible for the errors to be correlated.

We start by addressing the use of GRACE to estimate mass changes in the Arctic Archipelago. The "circularity" of the problem is explicitly mentioned in page 4, L32. We use an ensemble of ice sheet models and solid Earth rheologies to estimate the uncertainty in mass changes. Table 1 now gives the different combinations of ice-sheet and rheology models used to correct GRACE estimated mass changes. Four different ice sheet models and three different rheologies are used.

The ice sheet models correspond to two runs of the Glacial System Model for Northern Europe, the ICE-5G and the W12 models. As for the rheology we use the VM5a model and two models with an upper and lower mantle viscosities of  $16 \cdot 10^{20}$  and  $512 \cdot 10^{20}$  and  $10 \cdot 10^{20}$  Pas and  $100 \cdot 10^{20}$  Pas. We see that a weaker mantle leads to higher mass loss rates for the islands of the Artic Archipelago. We add this table as well as an explanation on the effect of the GIA model on the mass loss estimations in the tex.

GRACE data is not used in the creation of the OMCT ocean model but it is used in the ECCO ocean model. However, this is only 1 of the 40 data sets that are used to constrain the dynamic MITgcm ocean model and as shown in ECCO's documentationGRACE is one of the worst fitted observations. This fact is also evident in the results of Yu et al. 2018 who compare GRACE derived bottom pressure anomalies to the ECCO ocean model for the Argentine Gyre. We do not use the ECCO ocean model in our estimate itself, but only as a proxy of how much error we might expect from uncertainty in the ocean signal. Given the weak contribution of GRACE to the final output we think this is appropriate.

SED
The correlation of GRACE's measurement error and that from ice loss estimates cannot be ruled out. The error in ice loss estimates have two components (1) GRACE's accuracy error and (2) error due to uncertainty in the GIA model. While the second is not correlated with GRACE's measurement error, it is true that the first can be. In light of this discussion we decide to add the following discussion after equation (1).

"The assumption that errors are uncorrelated requires further discussion. GRACE data is assimilated in the ECCO ocean model. However, GRACE is only one of the 40 data sets used in the inversion process and the final product does not fit GRACE data well (Yu et al. 2018). Therefore there will be only a weak correlation with the GRACE data used in our estimation. Correlation between land surface hydrology models and present-day ice melt is not expected, because hydrology models have little skill in predicting trends and do not model areas of permanent snow. Finally, ice loss changes errors ( $\sigma_{ice}$ ) arise due to uncertainty in the GIA model and GRACE measurement error, we cannot rule out that the second error component might be correlated with  $\sigma_{GRACE}$ ."

In light of this concern, I would ask the authors to: 1) further quantify the effects of their processing technique for this area. In particular, by adding more discussion of the technique for idealized measurements in the context of the Barents; and 2) consider acknowledging the processed nature of these results by referring to them as "estimated gravity rates" rather than "observed gravity rates." I feel this is particularly important when the authors substitute the phrase "observed gravity rate" for the estimated maximum gravity rate (e.g., page 6 - line 31).

The processing techniques were detailed in the supplementary material of Root et al. (2015), their effect for an idealised measurement in the Barents Sea is shown there. Following the suggestions of the reviewer we: (1) Refer to the Supplementary material of Root et al. 2015 for a detailed explanation of how our processing affects an idealised signal (4,L5). (2) Include a Table showing the estimated mass loss changes obtained using different GIA models. (3) Use the term estimated gravity rates instead
**of observed (8,L14).**

The argument additionally suffers from another small, but troubling, circularity. The ensemble of ice models was chosen to represent two classes: empirical ice sheets developed using GIA observables and ice sheets developed from independent, process based models. However, all of the models are actually calibrated, in some way or another, to GIA observables with an implicit dependence on the assumed viscosity structure. For instance, the Tarasov samples are drawn from a distribution trained on GIA observables using the Peltier VM5a rheology. If the authors could comment on this bias and how that might account for the reference model being very near the best fit valley in all figures but the Siegert and Dowdeswell 2004 model, which is the only one to prefer an anomalously high viscosity, most likely because of it's earlier ice-free time.

Although the problem is certainly there, the S04, but also the UiT model are not fitted to GIA observations. In the main text we distinguish between two different types of ice models, (1) those that do not include ice sheet physics (ICE-5G, ICE-6G) and are entirely based on GIA observations and (2) those that incorporate ice sheet physics. However, as pointed out some of the ice models in the second subset do also include GIA observations. Tarasov's models are calibrated using the fit to RSL curves and uplift rates obtained with the VM5a model, however the calibration accounts for spread in the decay times due to uncertainty in the viscosity model and is not tuned to a single viscosity model as much as ICE-xG models are (L. Tarasov personal communication). For the UiT model a simple hydrostatic model is used to account for ice-elevation feedbacks, but the model fit to GIA observables is assessed a-posteriori (Patton et al. 2016). We include this additional information extending our description of the ice models in Section 2.2.

**Minor Comments**

page-line 4-33: "However, the GIA" It is not obvious that this should be so.
We tried to clarify this point by adding the individual mass loss estimates for each of GIA model in Table 1. Moreover, we clarify this point by showing that the uncertainty due to the GIA model used to recover the mass changes is of the same order of magnitude as GRACE's formal error (5,L7).

8-19: citing the  $\chi^2$  might make this point clearer. It is hard to tell that S04 is significantly worse than, say UiT, from figure 3.

We state that the model performs worse than the T1, T2 and T3 (which is clear from figure 3). Later on this is evidenced with the  $\chi^2$ . We also add some new discussion on the fit of the S04 model for different Earth models in Section 3.2 (9,L32).

10-32 might include "explicitly" in "not explicitly tied to a viscosity model"

We follow the suggestion

Figure 5 and Figure 6 - Could you note with a symbol the reference model and the best fit model in each of these plots?

For each lithospheric thickness we indicate the best fitting model with a red line and the reference model with a red dot.

- 1-2: in-> to "insight to the" Done
- 1-4: Split sentence Done
- 1-5 remove "a" in "a GIA models" Done

1-6 "is not negligible" and "should be taken into account" are redundant Done

1-16 Inconsistent use of "gravity disturbance rate" and "gravity rate" We use gravity disturbance rates until it is stated that the term gravity rates will be used instead in 4-L9.

3-7 missing word in "while best fitting models uplift rate measurements" Done

3-22 missing "and" in "GIA, and (post-) seismic" Done

SED
5-4 GAB undefined GAB is not an acronym. We add a reference to Flechtner et al. 2015 where the GAB files are defined.

5-9 "respectively" has no antecedent. Consider "both the OMCT and ECCO ocean models" We follow the suggestion

5-15 "while when" is difficult to parse We rephrase the sentence.

5-22 missing word in "This still allows" We rephrase accordingly.

5-23 correct citation parenthesis Done

5-25 missing "the" in "that of the unknown" Changed

5-26 remove nested parentheses Done

5-31 missing "the" in "the Earth's rheology" Added

8-12 I believe deglaciation starts earlier in T2 than in T1, unless I am much mistaken. That is true, we correct the misspelling.

8-31,8-32,9-30 "lower upper mantle viscosity" is pretty cumbersome to read. Consider something like "less viscous upper mantle" We follow the suggestion.

9-9 repeated word "which that" Error corrected

9-12 large->high "high upper mantle viscosity" We follow the suggestion.

9-16 typo"form" Error corrected

Figure 3 and Figure 4 - inconsistent x-axis label We modify the label to ensure consistency.

SED
120

10°

Fig. 1. Simulated GIA signal cut at different degrees
Fig. 2. Maximum gravity disturbance rate as function of cut-off degree

---

## Author Comment (AC2) · 3 Oct 2019

We thank the reviewer for the comments provided, we think they helped improving the quality of the manuscript and clarify some relevant points. Below, we address the reviewers' comments (in blue). A manuscript with the changes done to the manuscript shown in blue is also provided. References to pages and lines (page,L line) refer to the new version of manuscript, lines and pages mentioned in the reviewer's comments correspond to the first version of the manuscript. Figures in this response are referenced as R.

**Major comments**

1. My main concern is the lack of discussion of uncertainties in the resulting viscosities.

[Figure]

The conclusion that Fennoscandian upper mantle viscosity is a factor of 2 higher than that in the Barents Sea is given with very little discussion on uncertainties:

(a) Most studies would state resulting viscosities and elastic thickness as an interval determined one way or another from the statistics of the inversion process. A differently normalized chiËĘ2 range or a variance reduction, for example. On page 8, line 27, a 2 sigma interval is mentioned but not further referred to. The very different chiËĘ2 distributions for Barents Sea and Fennoscandia in Figures 5 and 6 make it difficult to asses which parts of the model space is appropriate to compare to one another. In addition, the color scale in the Figures does not enhance the well fitting regions very well, I suggest a scale with a better visible range.

We modify Figure 5 and 6 to give idea of the range of models that perform well. We define a confidence interval following Press et al. 1992 (Chapter 15) (8,L4). We indicate the values within the 95% confidence interval in grey. Additionally, to ease comparison, we indicate in red the best fitting model for each lithospheric thickness which eases the discussion in Section 3.2.

b) At least for the T1-T3 and S04 ice histories, even though the well fitting viscosity range starts at lower viscosities for the Barents Sea than for Fennoscandia, there is significant overlap at higher viscosity in Figure 5. This is less pronounced for ICE-XG and UiT in Figure 6, but is there at thicker elastic thickness. A more well defined range of which models are considered good fits would ease the comparison.

We agree there is an overlap between both regions. We chose to compare the best fitting upper mantle viscosity obtained for each lithospheric thickness in both regions (indicated in red in Figures 5 and 6) to illustrate the difference in upper mantle viscosity and elaborate more on the discussion (11,L11):

"We can infer lateral rheology changes by comparing the optimal Earth rheological parameters obtained for both regions. For each ice deglaciation chronology, we compare the two confidence intervals as well as the best fitting upper-mantle viscosity obtained

for each lithospheric thickness. We observe that for the UiT, ICE-5G and ICE-6G model both the confidence interval as well as the best fitting models have a systematically higher upper upper mantle viscosity in the Barents Sea as compared with Fennoscandia. This is also the case for the T1,T2 and T3 models when the best fitting models are compared, although there is an overlap of models of high upper mantle viscosity and thick lithospheres with a good fit in both regions. This systematic difference is likely evidence of lateral variation in Earth rheology. "

c) The lower bounds on viscosity is very similar for all ice models in the two regions. That is a little odd. Is there some bias somewhere? Such as they having similar Earth models during construction?

The lower bounds obtained for Fennoscandia and the Barents Sea are quite similar. This can be understood by studying the sensitivity of gravity disturbance rates to mass changes during different epochs (Figure R1). As upper mantle viscosity decreases the results are less sensitive to mass changes in the past where differences between ice models are more acute (i.e., Figure 1), making it more difficult to distinguish between ice sheet models. This agree with previous studies that have used gravity data to constrain solid Earth parameters. For instance, Steffen et al. 2010 finds similar lower bounds for upper mantle viscosity using the ICE-5G and the RSES models.

To illustrate this point we add the Figure R3 and add an explain along these lines in section 3.1 (10,L10). "The lower bound obtained with the other ice models is similar as models with a low viscosity have little sensitivity to mass changes during the early deglaciation phase, where differences between ice models are more manifest (Figure 2)"

d) Elastic thickness is discussed very briefly in the manuscript. The clear correlations in Figures 5 and 6 between viscosity and elastic thickness should be discussed further. This is different to the results in e.g. Steffen et al. (2010), Root et al. (2015a,2015b). How much are the results for a thicker elastic layer affected by the GRACE filtering pro-

cess? In addition, there are surely estimates from seismology of the (seismic) thickness of the lithosphere in the Barents and in Fennoscandia. These could perhaps also be used for comparison purposes, even though the measure a slightly different property.

The effect of lithospheric thickness is more evident than in other studies of GIA of the region. We compute the $\chi^2$ using the maximum gravity disturbance rates instead of averaging over a region as the shape of the signal in the Barents Sea is difficult to distinguish, for consistency we use the same strategy for Fennoscandia (Figure R2). A thinner lithosphere results in higher gravity rates and can counterbalance a low value of upper mantle viscosity. However, they also lead to narrow region of uplift which does not fit the signal well in Scandinavia. The effect of lithospheric thickness is also evident in Root et al. 2015a. An explanation along these lines is given in (10,L5)

We do not want to draw conclusions on the absolute values for the lithosphere thickness from our misfit plots, because we do not use the spatial pattern of the gravity rate (added in Section 3.1 10,L11) and this is why we compare best fitting upper mantle viscosity for each lithospheric thickness.

e) When concluding the factor of 2 viscosity difference between Barents Sea and Fennoscandia you should specify at which elastic thickness the comparison is made. If you use different elastic thickness for the different regions that should be explicit. Similarly for the seismic estimates of viscosity difference. These are at the same depth for the Barents Sea and Fennoscandia but to make a fair comparison it would be interesting with estimates of the seismic lithosphere thickness. How much does it matter for the comparison if there are differences in temperature and/or composition in the two regions?

As pointed out at b) the comparison is made for the best fitting upper mantle viscosity obtained for each lithospheric thickness. We add this explanation to the main text.

Differences in temperature are considered in the paper. With regard to composition, we now state that changes in composition are not considered but likely do not play as

large a role as temperature, in the upper mantle (11,L28-30).

f) You should compare your inferred viscosity differences to other GIA studies of the Barents Sea area and Fennoscandia. The large number of varying results for Fennoscandia indicate that such a comparison is non-trivial. For the Barents Sea, Root et al. (2015a) indicates $4 \cdot 10^{20}$ Pa s for the Barents and Auriac et al. (2016) has a very wide range of $2 - 20 \cdot 10^{20}$ Pa s.

We try to put our results in context by comparing them to the results of Auriac et al. (2016) which found a similar bound for upper mantle viscosity in the Barents Sea. For Fennoscandia we refer to the recent work of Simon et al. 2018 that give an overview of different rheology estimates.

Moreover, we are also more cautious in our claim about upper mantle viscosity constrain as other GRACE studies (Root et al. 2015b, Steffen et al. 2014) show that the effect of lower mantle viscosity is not negligible and obtain optimal fits for higher lower mantle viscosities than the one used in this study (6,L21). We add this discussion in section 3.2 and stress this fact in the conclusions.

2. The four error estimates for the GRACE processing seem very appropriate, but I wonder: (a) Is spherical harmonic degree 60 really enough for this study? And the filtering out of smaller wave-lengths seems to retain only very large scale features, on the order of the whole basin

See response given to the first comment of Reviewer 1.

b) The statement on page 5, line 12, about the independence of the estimates. It seems to me a little strange that the hydrological signal would be uncorrelated with the ice loss signal? Ice mass loss usually means melting, which surely influences the hydrology, both in time and magnitude. Is this not an issue?

See response given to the first comment of Reviewer 1. Moreover, land surface hydrology models have little skill in predicting trends and they do not contain permanent

snow or glacier therefore not much correlation with ice melt is expected. We add this explanation below equation 1.

c) The estimate of ice mass loss from GRACE data does indeed seem circular, and a little difficult. A GIA model using GRACE data is used to estimate uncertainties in the GRACE data for GIA applications? On page 4, lines 21-22 the authors state that the current ice mass changes "...partly mask the GIA signal...", but on lines 33-34 that "... the GIA model used to obtain the mass changes has a small effect in the recovered gravity rate trend...". This seems contradictory to me and need more detailed explanation. Do you use different GRACE filters here to capture the spatially smaller current deglaciations? Also, how are the error bars estimated from the range of ice and Earth models? Do you have a range of reasonable chiË{2} or something for this error estimate?

We add the mass loss changes obtained using the different GIA models in Table 1 as well as some more details about rheology and ice sheet model affects the results (see response Reviewer 1). What we mean with "... the GIA model used to obtain the mass changes has a small effect in the recovered gravity rate trend.." is that the error introduced by using different GIA is similar as that given by GRACE measurement error (as shown in Table R1). We agree that this statement is confusing and thus we change it by (5,L7): " The error in the derived mass changes due to uncertainty in GIA is similar to GRACE's measurement error".

**Minor comments**

Including GPS data from Svalbard and northern Norway would have been helpful to constrain the models. Why was this not done?

The focus of this work was on what can gravity tell us about interior structure in this two regions. We agree that adding GPS and even RSL might help to constrain the models but we decided to focus on the gravity signal because the gravity rate is available in the center of the Barents Sea in the region of largest ice thickness where other

measurements are absent.

Previous work (e.g., Auriac et al. 2016) have focused on GPS and RSL. To recognise the fact that we don't use these other data sets we add :

"We find that the ICE-5G, ICE-6G and UiT ice sheet models can be reconciled with GRACE observations provided the upper mantle viscosity is lower or the lithosphere thicker than in the VM2 model. The same conclusion is reached in Auriac et al. 2016 using GPS uplift measurements and RSL curves instead of gravity data."

It would be good to have a little bit more information on the ice reconstructions, especially with regard to the used Earth model physics for the non-GIA derived ice models. Do they have appropriate viscoelastic earths, or just simple hydrostatic adjustment, or...? Also, which time period do you use in the models? Just the deglaciation phase? If so, how are the ice sheets ramped up to the last glacial maximum?

We add more information about the different ice chronologies (see Section 2.2). We include both the build-up and deglaciation phase. We add (7,L21): "The only global ice sheet models are the ICE-5G and ICE-6G, for the other ice sheet models we use the ICE-6G ice model outside the EISC. We include the build-up and deglaciation phase of the last glacial cycle". Moreover, we add a new plot showing the deglaciation history in Fennoscandia (see Figure 2), and specify when the ice sheets start to build-up. We also include some information on how isostasy is implemented in the UiT model and S04 models.

Use regular non-italic font for units. Even in latex "Pa.s" can be made roman in math mode

We revise the text and make sure that Pa.s is non-italic.

1 3 You write insight into sub-surface structure. It is not really structure but rather rheology.

We follow the suggestion
2 30 Same as above

We follow the suggestion

1 4 Either spell out GRACE, or add "gravity" for clarity.

We follow the suggestion.

1 8 I would remove "deglaciation" here and describe the used time period in the paper.

For clarification we add "of the last glacial cycle" for clarification

1 16 Just to be clear, spell out GRACE or add gravity here the first time it is mentioned.

We follow the suggestion.

2 7 Here you could include the dynamic ice sheet model by Näslund et al. (2005): Näslund, J.-O., Jansson, P., Fastook, J. L., Johnson, J., and Andersson, L.: Detailed spatially distributed geothermal heat-flow data for modeling of basal temperatures and meltwater production beneath the Fennoscandian ice sheet, Ann. Glaciol., edited by: MacAyeal, D. R., International Glaciological Society, 40, 95– 101, doi:10.3189/172756405781813582, 2005.

We thank the reviewer for the suggestion but consider that the ice sheet models used in the study are enough to capture uncertainty in ice deglaciation chronology for the region.

3 7-8 "... best fitting models uplift rate measurements..." is difficult to understand. 3 29 "... we use ..." the software? There is an object missing in the sentence.

Indeed, we modify the sentence: "while best fitting models based on GPS uplift rate measurements have upper mantle viscosities"

4 6 Define "gravity disturbance rate" as opposed to "gravity anomaly rate".

Both terms are commonly used in physical geodesy, we add a reference to Hofmann-Wellenhof and Moritz physical geodesy book for clarification.

5 3 Reference to the ECCO model.

We add the reference to Forget et al. 2015 here.

5 4 What are the GAB products?

GAB files contain the ocean signal subtracted from GRACE and should be added back to restore GRACE's full ocean mass mass variations. We add a reference to Flechtner et al. 2015 for the GAB products.

5 8 No italics.

We follow the suggestion.

5 17 Ocean bottom pressure changes in the Baltic can be neglected? Are they so much smaller than in the Barents, or just relatively smaller?

Both, they are smaller in the Baltic Sea and the signal there is higher (see Figure R3), so we decided to not consider the error.

6 28-29 This sentence need reformulation.

We reformulate the sentence.

7 1 Are you using central Fennoscandia? If so, where is this?

We do not use a specific point in Fennoscandia but find the point with maximum rate.

7 9 In Figure 1 it is the gravity signal after processing, not necessarily the GIA signal.

The reviewer makes a valid point here. We reformulate the paragraph that now reads: "A clear positive anomaly is evident both in Fennoscandia and the central Barents Sea where the main domes of the Scandinavian Ice Sheet and Svalbard-Barents-Kara Ice Sheet were presumably located (Figure 1), we assume this signal to be entirely due to

GIA and call it the estimated GIA signal"

9 5 "A second set..." Which is the first?

We remove the $32 \cdot 10^{20}$ upper mantle viscosity from our plot as it corresponds to a higher viscosity than that used for the lower mantle. Doing so the second subset is less evident and thus we decide to remove this paragraph.

9 9 ".. gravity rate which that is larger than..." Fix this.

Done

9 20 The authors should point out that 3D effects are indeed significant, e.g. White-house et al. (2006), Steffen et al. (2006).

This is a good point. We modify the text accordingly which now reads. We decide to mention this in the method's section (6,L10-12)

" While this approach has been used in other GIA studies (Lambeck et al., 1998; Steffen et al., 2014), it has been suggested that far-field viscosity variations are relevant in Fennoscandia (Whitehouse et al. 2006)."

10 2 Which conclusion?

We add "of lateral viscosity changes between the two regions"

10 9 "...the reference model... a jump below 200 km" Please clarify which reference model and what the jump is, or refers to.

Changed to "a jump in the seismic velocity anomalies in the reference Earth models PREM and AK135

10 14 Stress for the flow law is taken from the GIA model. How accurate is this? Neglecting tectonics, topography, sediment loads etc surely distorts the "correct" stress state. How important is this?

Deviatoric stresses in the mantle from topography is small because of the long time-

scale. Stresses from sediments are small as current uplift rates are small (van der Wal and Ijpelaar 2017). We add to the text (11,L33) "Background stresses due to mantle convection are neglected as recent work suggest little interaction between GIA and mantle convection "

10 23 Why did you choose the 1500 m contour?

We choose this contour to encompass the Scandinavian landmass and the Barents Sea, but avoid areas where the ice thickness was thin. A sentence is now added on page Section 3.3 (12,L9)

10 31 You should define "significant", or rather add uncertainties.

We delete "significant".

12 1 "... the GRACE misfit"? The GRACE GIA models?

We change the sentence which now reads as: "This agrees very well with the results derived from the misfit of GIA models to GRACE data.

[Figure]

- $16 \cdot 10^{20}$ (Pa $\cdot$ s)
- $5 \cdot 10^{20}$ (Pa $\cdot$ s)
- $1 \cdot 10^{20}$ (Pa $\cdot$ s)

| $\mu_{UM}$ ($10^{20}$Pa $\cdot$ s) | $\tau$ (ka) |
|---|---|
| 32 | 12.6 |
| 16 | 10.2 |
| 8 | 8.8 |
| 5 | 7.5 |
| 3 | 4.5 |
| 1 | 2.6 |

**Fig. 1.** Gravity disturbance rates sensitivity to mass changes at a given epoch for different upper mantle viscosities. Relaxation times for different upper mantle viscosities is also given.

[Figure]

**Fig. 2.** Normalised \chi obtained using the maximum gravity rate (left) and averaging the \chi of points in Fennoscandia.

**Fig. 3.** Gravity disturbance rates due to ocean bottom pressure (from OMCT ocean model)

---

## Author Response (AR2)

**Reply to reviewer**

We thank the reviewer for her/his clear, constructive comments and summary these in the form of three suggestions. Below we consider the three suggestions made by the reviewer together with some of the comments they were based on. Changes to the original manuscript are indicated in blue. References within the text refer to the marked version of the manuscript.

The justification they give for cutting off the observations at l=60, is that higher order numbers contain negligible GIA signal. They support this assertion by showing the results of modeling GIA, using the ICE-5G model and an unspecified rheology. However, the paper is about fitting precisely these ice and rheological parameters to the GRACE observations. The authors have therefore pre-decided what the GIA signal should look like and which models should be considered valid.

1) Be very explicit about the decision to truncate at harmonic 60 and its possible consequences in this paper, not simply with reference to the Root supplemental material. Omitting that the decision is based on particular model runs is misleading at best.

In answering we will discuss both the truncation and Gaussian filtering as they have a similar effect, namely to reduce or remove the effect of noise in higher degree coefficients.

Indeed the argument we used to justify cutting at degree 60 and applying a lowpass filter was not satisfactorily explained in the paper and our previous rebuttal. It discussed at some length the effect of truncation and filtering on GIA models but only briefly mentioned that "We use a Gaussian filter to filter-out the noisy short wave-length gravity data and reduce GRACE's correlated errors which are evident as north-south stripes in the Level 2 data." while this is in fact the main reason to apply truncation and Gaussian smoothing.

Regarding the truncation we now add a reference to GRACE Technical Note 10 to explain that using coefficients higher degrees beyond 60 is not expected to lead to improve signal-to-noise (P3 L32).

We now add Figure 1a to figure 1 in the manuscript, which shows the gravity trends without filtering. It is evident that the signal is contaminated by noise. In figure 1c it can be seen that for 350 km Gaussian filter the signal is almost removed.

We now state that the lower value of the Gaussian filter halfwidth (200 km) is determined by the increase in noise in the GRACE data, while the higher limit (300 km) is determined by signal content (P4 L15-25).

We state that: *"All ice sheet models are sampled in a grid with a spatial resolution corresponding to a 128 degree Gaussian grid and the output of the model is truncated at degree 60 and processed using the same filters used to process the GRACE data"*. A similar statement is now added where the filters are first introduced (P4 15-17)

Thus the comparison between observations and modelled observations is done in a consistent way. We now acknowledge that the truncation and filtering means that sensitivity is inevitably lost (P4 L24-25).

The fits are biased toward long wavelengths by an additional methodological choice. The dataset the authors compile to fit maximum gravity disturbance rates is actually a single observation processed with filters of differing half widths - hardly a large set of unique information. It seems to me that the 200 km low-pass filter contains all the information of the 300 km low-pass filter, plus some smaller wavelength information. Thus, the lower frequency information (>300 km) is repeated in each data point, once again strongly biasing the fit to long-wavelength models, as with the decision to truncate.

As stated above we think that the Gaussian filtering is necessary to apply, but the exact value is somewhat arbitrary. The gravity rate as a function of wavelength should tell something, which is why we opted for investigating a range of half widths.

2)     Remove the formal claims of confidence regions, and describe the chi^2 values and the "data set" they are conditioned on as an appropriately approximate way to establish fit.

We agree that the way we constructed confidence regions was not rigorous and we thank the reviewer for pointing this out. We agree that the gravity rate for different filter half widths cannot be assume to be a set of uncorrelated observations and thus it becomes problematic to establish confidence intervals in which the 'true' Earth model is expected to lie, we removed formal claims regarding the confidence region and discuss the chi^2 values as a way to establish fit, following the suggestion.  Instead of highlighting misfit values above and below \chi^2=6 we now take the \chi^2=4 contour as separation, which corresponds to models deviating 2\sigma from GARCE's observations. We use this contour to define a subset of best fitting models that can not be distinguished from the best fit model. This explanation is added in Section 2.3. We updated the abstracts and conclusion; claims about confidence intervals and/or lower bound, have been replaced by statements about best fitting models which in the Barents Sea s have a lowest upper mantle viscosity of 3e20 Pas.

In particular, this approach strongly penalizes the S04 model and drives the rheology conditioned on it to the end of it's parameter range, perhaps artificially, causing me to remain skeptical of its fit.

3)     Include discussion about the possibility of short-wavelength components of the GIA gravity signal admitting different ice and rheology models.

Based on the discussion below point 1, we include in Section 3.1 that our decision of filtering shortwave lengths leads to a loss of sensitivity  and acknowledge that details in ice history can not be assessed by our models (P4 L24-25).

However, we think this does not mean that misfit is biased against high-wavelength models (such as presumable model S04), only that small-scale features in the models can not be seen in the data. To support this point we show the effect of filtering the signal of different GIA models (Figure 2). The filter reduces the maximum gravity signal in the region and this reduction is more or less pronounced depending on the ice and Earth model. If we compare model T1 with a lithosphere of 40km with ICE-6G with the reference Earth model we see that gravity disturbance rates are different for the case without filtering but get very close when applying a 300km Gaussian filter.

We do not think that this approach unfairly penalises the S04 model. In Figure 3 of this rebuttal we see that this model does not particularly show more higher wavelength features, but mainly has much less ice in the Barents Sea and Fennoscandia and becomes ice-free much earlier compared to the other ice sheet models (figure 2 of the manuscript). This means that unless a very high upper mantle viscosity is used, most of the solid Earth relaxation has already occurred (see Sensitivity plot Figure 5 in the old manuscript) which results in small gravity disturbance rates in the region as compared with the other models (as shown in the bottom row Figure 3). This point is added to the text on P11 L5-6. This result is consistent with previous studies that have used other data sets (Auriac et al. 2016) and is not particular for our processing approach. To further illustrate the differences between ice sheet models we add in Figure 2 of the manuscript ice sheet thickness and extent for the most massive (ICE-6G) and least massive (S04) ice sheet models for two different time epochs.

[Figure]

Figure 1: Gravity disturbance rates in the central Barents Sea without filtering (a), with a 200km filter (b) and with a 350 km filter (c).

[Figure]

Figure 2: Maximum gravity disturbance rates obtained with different ice deglaciation chronologies and two different Earth models: the reference Earth model (solid lines) and an Earth model with upper mantle viscosity equal to 16e20 Pa s and a 40km lithosphere.

[Figure]

Figure 3: Ice thickness for 25ka BP and 15ka BP and resulting unfiltered gravity disturbances rates for the ICE-6G (left), T1 (center) and the S04 (right) models

[revised manuscript text omitted]